# Cell metabolism regulates integrin mechanosensing via an SLC3A2-dependent sphingolipid biosynthesis pathway

Etienne Boulter [1], Soline Estrach [1], Floriane S. Tissot [1], Marco L. Hennrich [2,3], Lionel Tosello[1], Laurence Cailleteau[1], Laura R. de la Ballina [1], Sabrina Pisano [1], Anne-Claude Gavin [2,3] & Chloé C. Féral [1]

Mechanical and metabolic cues independently contribute to the regulation of cell and tissue homeostasis. However, how they cross-regulate each other during this process remains largely unknown. Here, we show that cellular metabolism can regulate integrin rigidity-sensing via the sphingolipid metabolic pathway controlled by the amino acid transporter and integrin coreceptor CD98hc (SLC3A2). Genetic invalidation of CD98hc in dermal cells and tissue impairs rigidity sensing and mechanical signaling downstream of integrins, including RhoA activation, resulting in aberrant tissue mechanical homeostasis. Unexpectedly, we found that this regulation does not occur directly through regulation of integrins by CD98hc but indirectly, via the regulation of sphingolipid synthesis and the delta-4-desaturase DES2. Loss of CD98hc decreases sphingolipid availability preventing proper membrane recruitment, shuttling and activation of upstream regulators of RhoA including Src kinases and GEF-H1. Altogether, our results unravel a novel cross-talk regulation between integrin mechanosensing and cellular metabolism which may constitute an important new regulatory framework contributing to mechanical homeostasis.

---

[1] Institut National de la Santé et de la Recherche Médicale (INSERM) U1081, Centre National de la Recherche Scientifique UMR 7284, Université Cote d'Azur, Institute for Research on Cancer and Aging, Nice (IRCAN), Nice 06107, France. [2] European Molecular Biology Laboratory (EMBL), Structural and Computational Biology Unit, Meyerhofstraße 1, D69117 Heidelberg, Germany. [3] Molecular Medicine Partnership Unit (MMPU), Meyerhofstraße 1, D69117 Heidelberg, Germany. These authors contributed equally: Etienne Boulter, Soline Estrach. Correspondence and requests for materials should be addressed to E.B. (email: etienne.boulter@inserm.fr) or to C.C.F. (email: chloe.feral@inserm.fr)

Many factors influence cell behavior and tissue homeostasis. Among those, mechanical signals are particularly interesting since they are becoming increasingly prominent in the regulation of many physiological processes including development and morphogenesis as well as in several pathological conditions such as atherosclerosis or cancer[1–3]. In a solid tissue, most of these mechanical constraints arise from the interactions with neighboring cells and with the extracellular matrix (ECM), and must constantly be monitored. However, unlike classical chemical signals, mechanical forces have to be converted into a chemical signal for the purpose of intracellular signaling[4]. Such a conversion process occurs in several structures in cells including cell–matrix adhesion complexes, which are organized around receptors of the integrin family bound to an actin-coupled intracellular complex[5]. Central to this complex, integrins are heterodimeric receptors devoid of catalytic activity that can function as classical ECM receptors but also as mechanosensors, conveying externally applied forces to the intracellular complex[6]. Some adhesion complex proteins such as talin or kindlins control the activation state of the integrin receptor while others affect integrin function in a more elusive manner such as the integrin coreceptor CD98hc (SLC3A2), which does not affect integrin activation[7].

CD98hc is a bifunctional protein that serves as a regulatory subunit of the heteromeric amino acid transporter (HAT) system[8] and simultaneously as an integrin coreceptor[9]. CD98hc is a single span type II transmembrane protein that associates with one of several SLC7 light chains via its extracellular domain and with integrins β1 and β3 via its transmembrane and intracellular domains[10]. The HATs function as exchangers which selectively transport large cationic, neutral, small neutral, and negatively charged amino acid[11]. On the integrin side, CD98hc regulates signaling downstream of integrin engagement including FAK, Akt, and Src phosphorylation, as well as Rac1 activity and integrin-dependent processes such as matrix assembly, cell proliferation and tumor growth[7,12,13].

Therefore, CD98hc lies at the crossroads between integrins and amino acid transporters, or from a conceptual standpoint, between integrin function and cell metabolism. This physical and functional connection is gripping since cross-talk regulation between integrins and cell metabolism is emerging as a novel paradigm in the regulation of cell behavior[14]. Recent reports indicate that integrins regulate critical controllers of cell metabolism such as AMPK or mTOR[14,15] as well as effector intermediates such as metabolite transporters[16]. Indeed, in the pathological context of cancer, intricate and bidirectional relationships connect integrins and cell metabolism, governing both over integrin expression and function as well as over cell metabolism[14]. Interestingly, this regulation may be extended and generalized to other types of adhesion receptors such as E-cadherin which regulates cell metabolism through AMPK[17].

Our recent findings indicate that CD98hc regulates ras-driven tumorigenesis by modulating integrin-mediated mechanotransduction[18]. This seemingly suggests that CD98hc may regulate integrin-mediated mechanosensing, on top of classical integrin signaling, which has never been formally addressed. Therefore, while an interplay between classical integrin engagement and signaling, and several key components of cell metabolism exists, it is still unclear if and how integrin mechanical signaling and cell metabolism can regulate each other and how this may affect cell and tissue behavior.

Here, we show that cellular metabolism can regulate integrin rigidity sensing via the sphingolipid metabolism controlled by the amino acid transporter and integrin coreceptor CD98hc (SLC3A2). We show that depletion of CD98hc in cells impairs rigidity sensing and mechanical signaling downstream of integrins, including RhoA activation. In mice, genetic deletion of CD98hc in dermal fibroblasts results in aberrant tissue mechanical homeostasis including defective ECM assembly. At the molecular level, we found that CD98hc controls sphingolipid biosynthesis via the regulation of the level of the delta-4-desaturase DES2. Loss of CD98hc decreases DES2 levels and sphingolipid availability, which prevents proper membrane recruitment, shuttling and activation of upstream regulators of RhoA such as Src kinases and GEF-H1, and ultimately, rigidity sensing possibly via alteration of membrane microdomains. Altogether, our results reveal an unexpected regulation of integrin mechanosensing by cellular metabolism. This mechanism suggests a new regulatory framework between integrin mechanosensing and cell metabolism which questions many biological situations or pathological conditions associated with metabolic changes and altered mechanical properties.

## Results

**CD98hc regulates integrin mechanosensing and rigidity sensing.** CD98hc (SLC3A2) has long been associated with the regulation of integrin function[7,9]. Among the variety of integrin-regulated processes, we recently found that CD98hc was linked to stiffness-dependent skin tumor progression[18] suggesting that CD98hc may regulate integrin-mediated mechanosensing. To explore this possibility, we isolated and immortalized dermal fibroblasts from CD98hc dermal conditional *Fsp1-Cre, CD98hc*fl/fl knock-out mice (further referred to as KO or CD98hc-null cells) and control *CD98hc*fl/fl littermates (referred to as control cells). *Fsp1-Cre, CD98hc*fl/fl mice were viable, fertile and displayed specific deletion of CD98hc in the dermal compartment (Supplementary Fig. 1a, b). As expected, KO fibroblasts did not express CD98hc (Supplementary Fig. 1c) but did express characteristic fibroblast markers such as vimentin (Supplementary Fig. 1d, e). One typical cellular response to mechanical force application on integrins includes RhoA-driven cell stiffening[19]. Therefore, in order to assess whether CD98hc regulates integrin mechanosensing, we monitored cell stiffening using fibronectin (FN)-coated paramagnetic beads coupled to magnetic tweezers to sequentially apply forces on cells and assess their stiffening by tracking bead movement[19]. We applied a regimen of five force pulses of 8.2 pN of magnitude (±0.49 pN) for 4 s followed by 6 s of rest (Fig. 1a). In control cells, relative bead displacement decreased for each pulse (Fig. 1a, b) demonstrating that, as previously described[19], cells stiffened upon force application on integrins. On the contrary, depletion of CD98hc prevented cell stiffening (Fig. 1b) indicating that CD98hc is required for proper cellular response to mechanical force application on integrins. To determine whether this resulted from defective mechanical sensing or defective cellular response, we applied mechanical forces on integrin receptors using the same beads, and monitored RhoA activation upon loss of CD98hc[19]. In control cells, force application triggered RhoA activation as early as 1 min whereas it barely stimulated RhoA in CD98hc-null cells (Fig. 1c and Supplementary Fig. 1j). Depletion of CD98hc in HeLa cells by siRNA produced similar effects (Supplementary Fig. 1g, h) highlighting the universality of this regulatory mechanism. As a control, force-dependent RhoA activation could be rescued in CD98hc-null cells by re-expressing wild-type CD98hc (Fig. 1d and Supplementary Fig. 1k). In contrast, stimulation of RhoA by serum, a classical RhoA agonist[20], was not affected by loss of CD98hc (Fig. 1f), as previously reported for lysophosphatidic acid stimulation in mouse embryo fibroblasts[12], indicating this regulation is specific to mechanical signals. Activation of Rac1, another mechanoregulated Rho family member[21], was neither triggered by force application nor influenced by loss of CD98hc (Fig. 1g) highlighting the specificity

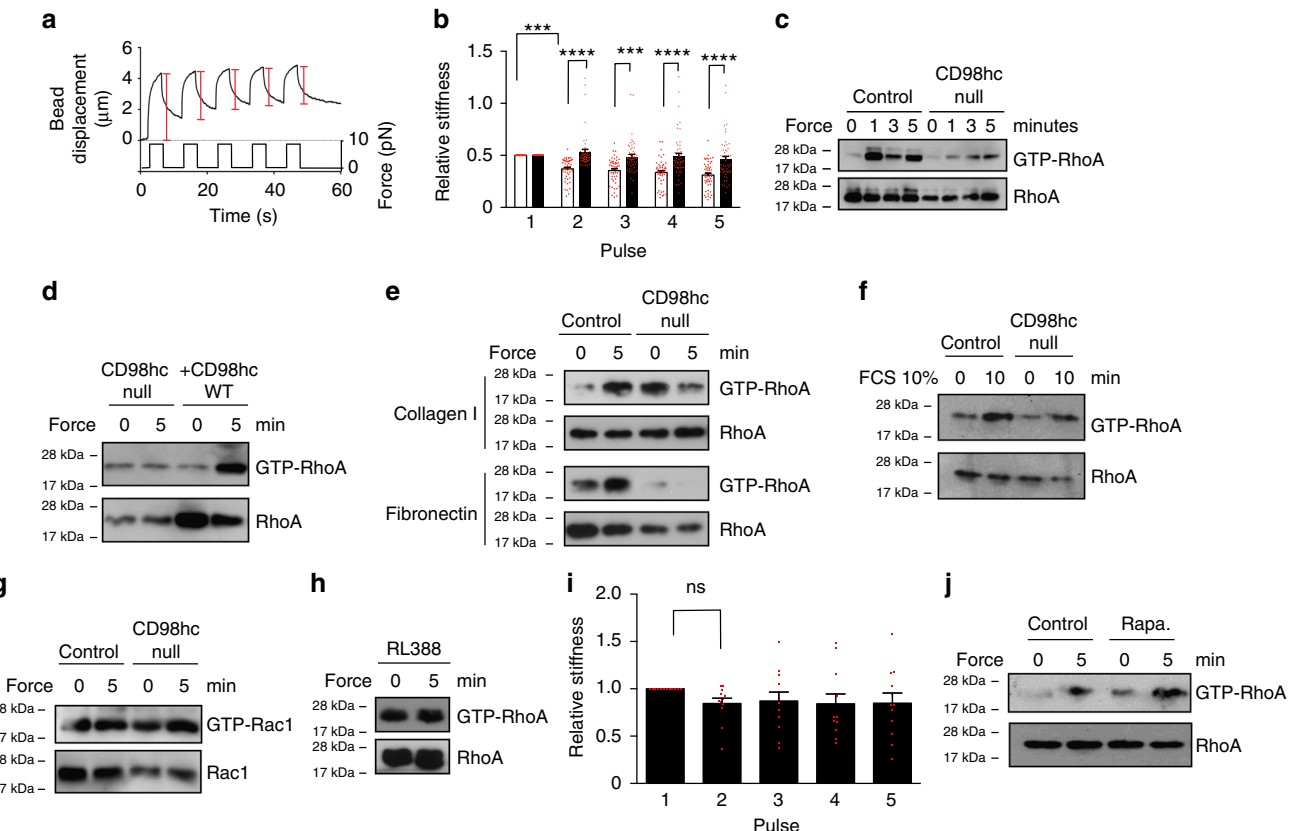

**Fig. 1** CD98hc depletion impairs RhoA activation and cell response to mechanical force application on integrins. **a**, **b** Loss of CD98hc impairs cell stiffening triggered by successive pulses of mechanical force applied to integrins. Red brackets, maximal bead displacement shown for a control experiment. (Mean + s.e.m., $n = 2$ with at least 25 beads each, ***$P < 0.001$ ****$P < 0.0001$ in a two-way ANOVA). **c** Genetic deletion of CD98hc in dermal fibroblasts impairs RhoA activation triggered by mechanical force application on integrins using fibronectin (FN)-coated paramagnetic beads, one experiment representative of $n = 2$. **d** Re-expression of WT CD98hc in CD98hc-null fibroblasts rescues mechanically coupled RhoA activation, one experiment representative of $n = 3$. **e** Genetic deletion of CD98hc in dermal fibroblasts impairs RhoA activation triggered by mechanical force application on integrins using either collagen I (ColI)-coated or FN-coated paramagnetic beads, one experiment representative of $n = 3$. **f** Loss of CD98hc in fibroblasts does not affect FCS-induced RhoA activation, one experiment representative of $n = 2$. **g** Loss of CD98hc in fibroblasts does not affect Rac1 regulation by mechanical forces. **h**, **i** Direct mechanical stimulation of CD98hc with RL388-coated magnetic beads does not trigger RhoA activation nor does it stimulate cell stiffening. **i** Means are plotted with s.e.m. as error bars from $n = 12$ beads, statistical analysis was performed in a two-way ANOVA. **j** mTORC1 inhibition with rapamycin 100 μM for 1 h does not impair mechanically coupled RhoA activation using FN-coated magnetic beads, one experiment representative of $n = 2$

of this regulation to RhoA. Also, using type I collagen-coated beads, an abundant dermal ECM protein, resulted in the same effects on RhoA activation as FN (Fig. 1e and Supplementary Fig. 1l) highlighting the physiological relevance of this regulation and indicating that CD98hc may regulate mechanosignaling downstream of integrin β1. Along this line, cell surface expression of integrin β1, β3, αv, α1, and α5 subunits were comparable in both cells (Supplementary Fig. 1f) and neither integrins α2, α10, nor α11 were detected. The amount of conformationally active integrin β1 was not affected by loss of CD98hc (Supplementary Fig. 1f). Also, we applied forces directly on CD98hc using anti-CD98hc antibody-coated beads, which induced neither RhoA activation nor cell stiffening (Fig. 1h, i) indicating that CD98hc is not per se a mechanosensor. Finally, CD98hc-dependent amino acid transport may affect activation of mTOR[22], which has been reported to influence integrin function[14]. Here, inhibition of mTORC1 did not affect mechanically coupled RhoA activation, ruling out mTORC1 as a potential regulator of this process (Fig. 1j).

To assess if loss of CD98hc would also affect rigidity sensing, we grew cells for 12 h on FN-coated hydrogels of different elastic modulus ($E$) ranging from 0.5 to 50 kPa. We observed that while control cells spread properly already when $E$ is low (8 kPa),

CD98hc-null cells remained rounded until $E$ reaches a value of 50 kPa (Fig. 2a). At the molecular level, increasing hydrogel rigidity induced RhoA activation only in control fibroblasts (Fig. 2b). On physiologically relevant 8 kPa FN-coated hydrogels, control cells generated forces of higher magnitude than that of CD98hc-null cells (Fig. 2c) as measured by traction force microscopy (TFM). Increasing substrate stiffness from 0.5 to 8 kPa triggered a contractile response in control cells without equivalent counterpart in CD98hc-null cells (Fig. 2d). Forcing RhoA activation with CNF1 in CD98hc-null cells on 8 kPa hydrogels restored force magnitude to levels similar to that of control cells (Supplementary Fig. 2a) indicating that cell contractility per se is not defective in these cells. Additionally, we show that loss of CD98hc impairs stiffness-dependent activation of YAP/Taz[23]. In control cells, increased gel stiffness induced activation of YAP/Taz as monitored by reverse transcription quantitative polymerase chain reaction (RT-qPCR) of ANKRD1 and CTGF as well as nuclear translocation of YAP (Fig. 2e and Supplementary Fig. 2b, c). In contrast, in CD98hc-null cells, YAP/Taz activity always remained basal (Fig. 2e) and YAP mostly cytosolic (Supplementary Fig. 2b). This also could be rescued by forcing RhoA activation (Fig. 2f). Altogether, this indicates that CD98hc is essential to sense and generate a proper

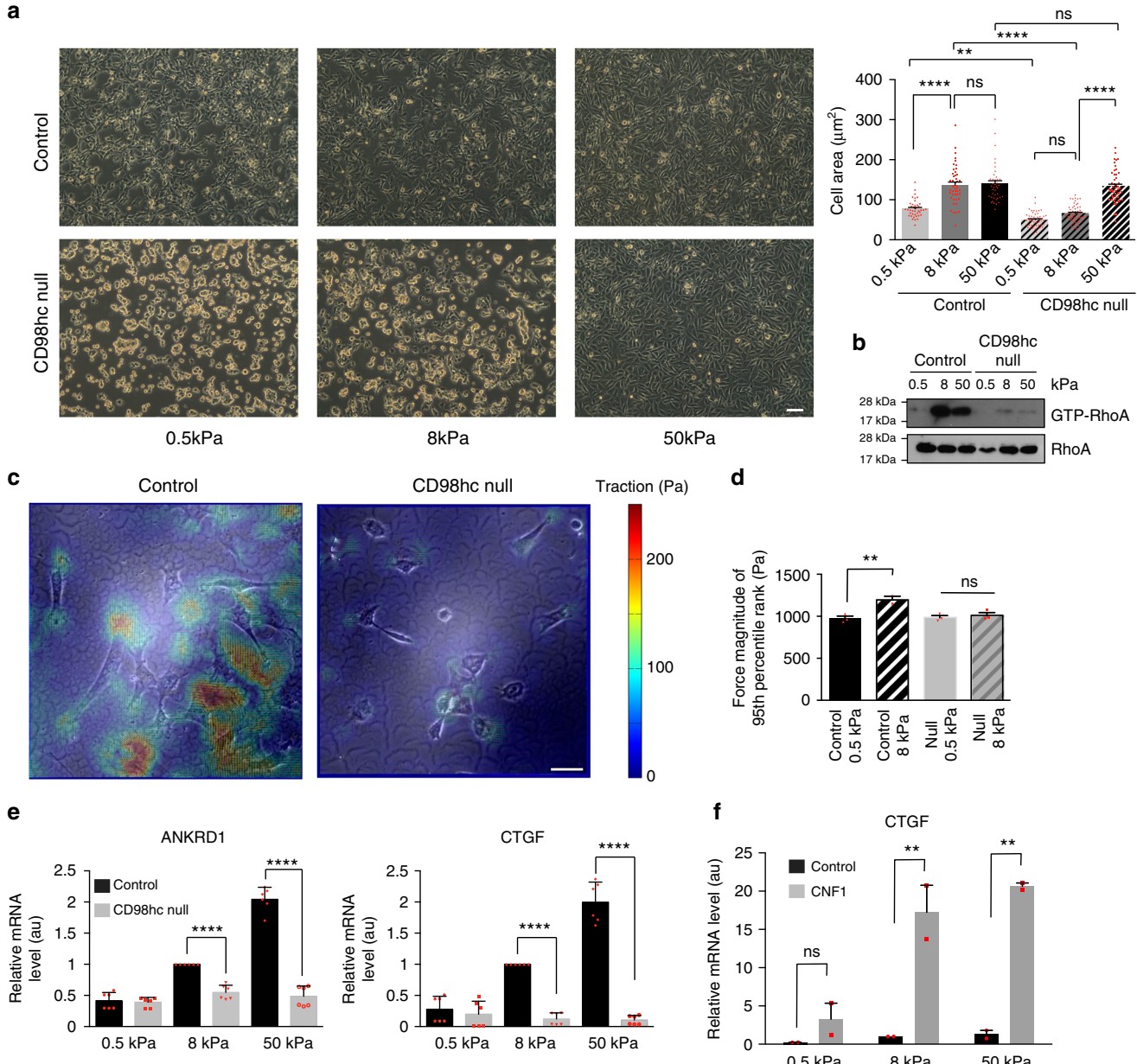

**Fig. 2** CD98hc depletion impairs rigidity sensing. **a** Representative morphology of control or CD98hc-null dermal fibroblasts grown on FN-coated hydrogels of specified elastic modulus. Scale bar is 50 μm. Cell area is plotted as means with s.e.m. as error bars ($n = 40–54$ cells, $^{**}P < 0.01$, $^{****}P < 0.0001$ in a one-way ANOVA). **b** Loss of CD98hc in fibroblasts impairs RhoA activation upon substrate stiffening. **c, d** Loss of CD98hc impairs cell contractility (**c**) and rigidity sensing (**d**). Traction force microscopy was performed on subconfluent cells. Black and black hatched bar, control cells, gray and gray hatched bar, CD98hc-null cells (mean + s.e.m., $n = 3$, $^{**}P < 0.01$ in a one-way ANOVA). Scale bar is 50 μm. **e** Loss of CD98hc impairs YAP/Taz activation by mechanical cues as measured by ANKRD1 and CTGF transcript quantification by RT-qPCR. Black bars, control cells and gray bars, CD98hc-null cells (mean + s.e.m., $n = 3$, $^{****}P < 0.0001$ in a Student's $t$ test). **f** Forcing RhoA activation using CNF1 in CD98hc-null cells rescues YAP/Taz activation by mechanical cues. Means are plotted with s.e.m. as error bars. $n = 2$ $^{**}P < 0.01$ in a Student's $t$ test

cellular response to mechanical force application and perform rigidity sensing. At the molecular level, CD98hc is required for mechanical signaling to RhoA.

**Loss of CD98hc compromises tissue mechanical homeostasis.** Based on these results, we envisioned that loss of CD98hc in dermal fibroblasts, in vivo, may result in altered mechanical signaling and tissue mechanical homeostasis. Indeed, in *Fsp1-Cre*, *CD98hc*^fl/fl mice, growing hair follicles behaved abnormally as

compared to *CD98hc*^fl/fl mice (Fig. 3a) suggesting that environmental conditions, such as mechanical properties, may be altered. Upon loss of CD98hc, *E* of dermal tissue decreased from 3.90 kPa in *CD98hc*^fl/fl to 0.91 kPa in *Fsp1Cre*, *CD98hc*^fl/fl mice (Fig. 3b and Supplementary Fig. 3a). At the ultrastructure level, local zones of high-*E* values were observed only in control dermis (Fig. 3c). Besides, 3D topographic reconstruction showed poorly structured *Fsp1Cre*, *CD98hc*^fl/fl dermis (Supplementary Fig. 3b) suggesting that ECM topography and organization may be altered. Indeed, while collagen deposition was unaffected in *CD98hc*^fl/fl mice

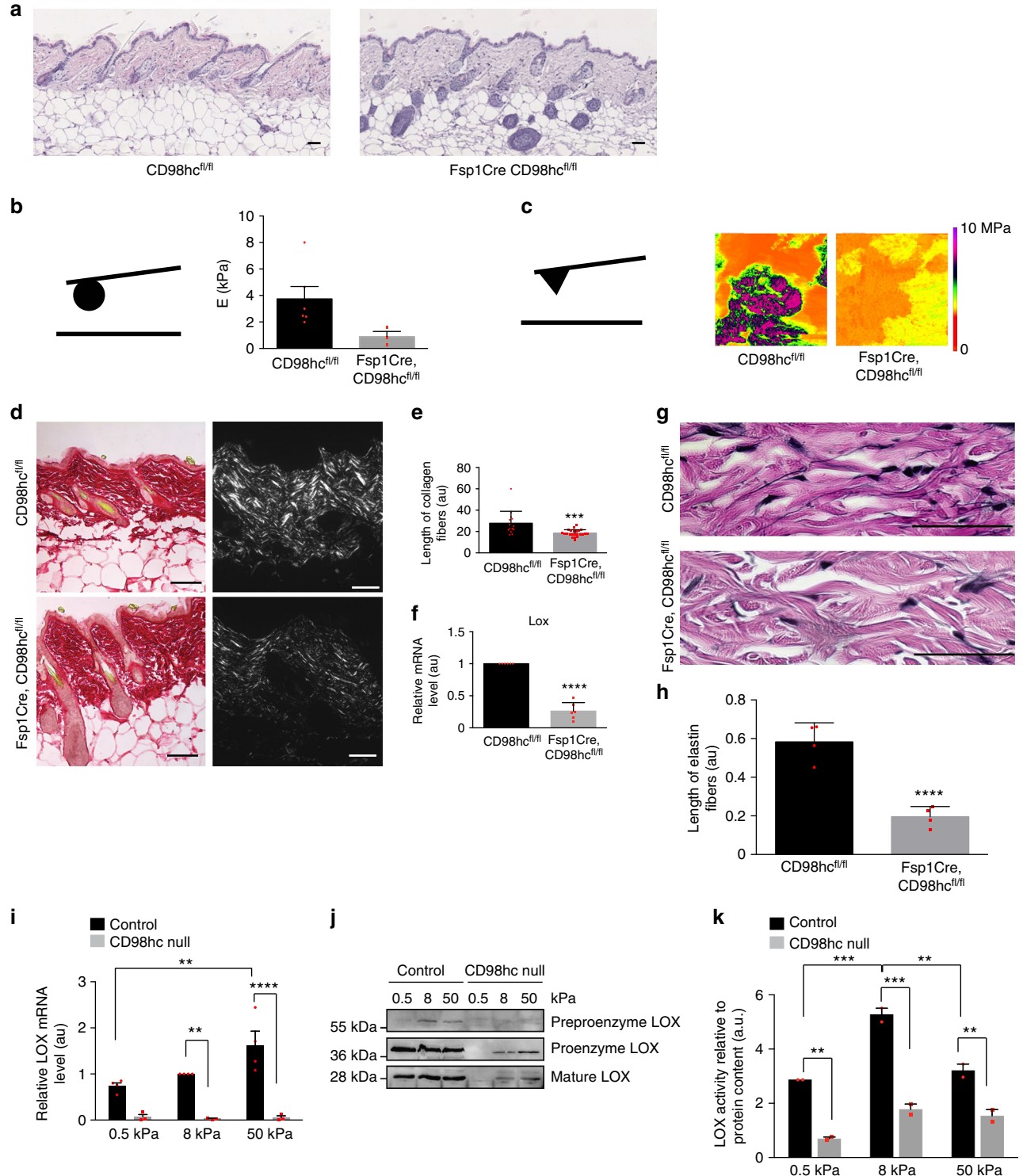

(Fig. 3d), both collagen and elastin fiber assembly were altered in *Fsp1Cre, CD98hc*^fl/fl mice (Fig. 3d, e, g, h). We correlated that faulty collagen assembly to a reduced expression of collagen cross-linking enzymes LOX, LOXL1, and TG2 (Fig. 3f and Supplementary Fig. 3c, d). In vitro LOX mRNA and protein expression, as well as its catalytic activity, were stiffness-dependent in control fibroblasts and impaired upon loss of CD98hc (Fig. 3i–k). Thus, CD98hc is required for proper tissue mechanical homeostasis thanks to proper mechanically coupled ECM organization.

**CD98hc regulates de novo sphingolipid biosynthesis via DES2.** From our results, it emerges that CD98hc is essential for mechanosignaling and mechanosensing downstream of integrins. However, how this occurs at the molecular level remained elusive and prompted us to investigate further this molecular regulation. Besides integrin signaling[9], CD98hc regulates a variety of cellular functions including amino acid transport[8] or cell metabolism in a broader view. We envisioned that some of those functions may indeed cross-regulate integrin signaling and mechanosensing. Therefore, we screened various CD98hc mutants and chimeras,

**Fig. 3** CD98hc is required for proper skin mechanical homeostasis in vivo. **a** Representative skin sagittal sections of CD98hcfl/fl or Fsp1Cre, CD98hcfl/fl mice stained with H&E. Scale bar is 50 μm. **b** Genetic deletion of CD98hc in dermal fibroblasts alters dermal mechanical properties. Young's elastic modulus ($E$) of the dermis was measured by AFM with a spherical probe (mean with s.d. as error bars, $n = 5$ and 4 mice, respectively, $^*P < 0.05$ in a Student's $t$ test). **c** $E$ magnitude heatmaps of areas of dermis scanned by AFM using a pyramidal probe. **d, e** Loss of CD98hc in dermal fibroblasts reduces collagen assembly as observed by picrosirius red staining under transmitted or polarized light (**d**), fiber length (**e**) (mean with s.d. as error bars, at least 300 fibers were counted with $n = 3$ $^{**}P < 0.01$ in a Welch's $t$ test (**e**)). **f** Relative mRNA levels of LOX was measured by RT-qPCR on total mRNA extracted from skin of CD98hcfl/fl or Fsp1Cre, CD98hcfl/fl mice (mean with s.d. as error bars, $n = 6$ $^{****}P < 0.0001$ in a Student's $t$ test). **g, h** Loss of CD98hc in dermal fibroblasts impairs elastin fibers assembly as observed on skin sagittal sections stained for elastin. Scale bar is 50 μm. Means are plotted with s.d. as error bars. At least 300 fibers were measured on sections from $n = 3$ mice. $^{****}P < 0.0001$ in a Student's $t$ test. **i** Relative mRNA levels of LOX was measured by RT-qPCR on total mRNA extracted from control or CD98hc-null dermal fibroblasts grown FN-coated hydrogels of indicated stiffness for 24 h. $n = 2$, means are plotted with s.e.m. as error bars, $^{**}P < 0.01$, $^{****}P < 0.0001$ in a Student's $t$ test. **j** Total cell lysates from control or CD98hc-null dermal fibroblasts grown FN-coated hydrogels of indicated stiffness for 24 h were resolved by SDS–PAGE and analyzed by Western blotting. **k** Relative LOX activity was measured in cell lysates from control or CD98hc-null dermal fibroblasts grown FN-coated hydrogels of indicated stiffness for 24 h. $n = 2$, means are plotted with s.e.m. as error bars. $^{**}P < 0.01$, $^{***}P < 0.001$ in a two-way ANOVA

---

expressed in a CD98hc-null background, for simultaneous defective mechanosensing and altered metabolic profile. Practically, we monitored RhoA activation upon application of mechanical forces on integrins and performed metabolomic profiling for each construct (Fig. 4a). For each construct, we ran biological replicates and only metabolites consistently affected were further considered (see Methods). In CD98hc-null cells, 238 metabolites out of a total of 447 were altered as compared to control cells (Student's $t$ test, $P < 0.01$, Benjamini–Hochberg [BH] procedure FDR < 5%, $n = 4$). These metabolites belonged to a wide variety of metabolic pathways, and included, as expected, amino acid metabolism, suggesting that the loss of CD98hc function broadly affects metabolism (Supplementary Fig. 4a). Among all mutants and chimeras tested, only C330S stood out as it failed to rescue mechanosensing in CD98hc-null cells (Fig. 4b, c, and Supplementary Fig. 4b–d) and simultaneously displayed a much cleaner metabolic signature as only 72 out of 447 metabolites were altered (Fig. 4d) (Student's $t$ test, $P < 0.01$, BH procedure FDR < 5%, $n = 4$). Among them, sphingolipid metabolism showed an intriguing pattern (Fig. 4d, Fig. 5i, Supplementary Table 1 and Supplementary Data 1). Out of the 14 sphingomyelins and the 3 glycosphingolipids that were analyzed, 5 were downregulated (Student's $t$ test, $P < 0.01$, BH procedure with FDR $q < 5\%$, $n = 4$) (Fig. 4d) while the metabolites of the initial de novo synthesis pathway (3-ketosphingosine, sphinganine, and N-palmitoyl-sphinganine) were all upregulated (Student's $t$ test, $P < 0.01$, BH procedure FDR < 5%, $n = 4$) (Figs. 4d, 5i). As a control, the amino acid transport defective C109S mutant[24] rescued RhoA activation after mechanical stress and had opposite effects on sphingolipid metabolism, indicating that CD98hc-dependent amino acid transport did not regulate mechanosensing (Fig. 4b, c, and Supplementary Data 1).

Sphingolipids are membrane constituents that participate to the formation of the myelin sheath in neurons and, together with cholesterol, to that of membrane microdomains in cells, including cholesterol-enriched membrane microdomains (CEMM) also formerly known as lipid rafts[25]. Equally important, a number of sphingolipids are bioactive lipids that behave as cellular second messengers including ceramides, sphingosine and sphingosine-1-phosphate (S1P) to name a few[26].

This reduction of sphingolipid levels, in CD98hc-null or C330S-expressing cells, suggests that a specific sphingolipid-enriched compartment and/or sphingolipid second messengers may regulate mechanosensing. Unfortunately, the wide range and complexity of sphingolipid second messengers hinders the possibility to reasonably assess each one individually. Nevertheless, among those, we found that inhibition of S1P, previously associated with RhoA activation[27], did not prevent mechanically coupled RhoA activation (Fig. 4e). Sphingolipids may also regulate RhoA activation via membrane microdomains such as

CEMM. Supporting this hypothesis, we observed a reduction in cholesterol levels as well as in sphingomyelin levels in CD98hc-null cells (Supplementary Figs. 4f, g, 8c). Indeed, cholesterol depletion or sphingolipid synthesis inhibition in control cells resulted in impaired RhoA activation in response to force application on integrins confirming that sphingolipids and cholesterol are required for proper integrin mechanosensing (Fig. 4f). Conversely, exogenous supply of sphingomyelin in C330S-expressing cells was sufficient to restore RhoA activation by mechanical forces and mechanically coupled gene transcription indicating that sphingolipid depletion is the sole defect preventing proper mechanosensing in C330S-expressing cells (Fig. 4g, h). Interestingly, exogenous supply of sphingomyelin did not rescue RhoA activation in CD98hc-null cells (Supplementary Fig. 4e) indicating that not only proper sphingolipid synthesis is required for integrin mechanosensing but that CD98hc also has additional functions. This was further supported by expression of two function-defective chimeras of CD98hc in KO fibroblasts, C98T98E69 and C69T69E98 (Supplementary Table 2). Both chimeras failed to rescue mechanosensing indicating that regulation of integrin mechanosensing by CD98hc requires both integrin binding (C69T69E98) as well as integrin-independent functions controlled by C330 (C98T98E69) (Supplementary Fig. 4h, i). In both cases, numerous metabolic alterations could be observed including in sphingolipid biosynthesis (Supplementary Fig. 4j, k, Supplementary Table 1 and Supplementary Data 1).

We next wondered how CD98hc could regulate sphingolipid biosynthesis. Quantification and analysis of metabolites along the de novo sphingolipid biosynthesis pathway in C330S-expressing cells revealed that metabolites up to ceramide synthesis were upregulated while downstream of that enzymatic step, they were downregulated (Fig. 5i and Supplementary Table 1). This suggested that sphingolipid depletion may result from inhibition of the delta-4-desaturase enzymes DES1 and DES2. While their respective specific activities are still debated, DES1 is generally ubiquitously expressed while DES2 is preferentially expressed in specific tissues including skin where it prevails over DES1[28]. In CD98hc-null cells, we found that while DES1 expression was fourfold upregulated, DES2 expression was reduced by 70% (Fig. 5a, b, and Supplementary Fig. 5a, b). A similar trend was observed in C330S-expressing cells since DES2 expression was also reduced of 70%, while DES1 expression tended to increase (Fig. 5c, d, and Supplementary Fig. 5c, d). For both enzymes, expression of WT CD98hc in null cells would restore initial enzyme level (Supplementary Fig. 5e, f). Since transcriptional mechanisms did not account for the regulation of the expression of DES2 (Fig. 5e–h), we hypothesized that it may be linked to its stability or degradation. None of the degradation pathways we tested seemed to account for its downregulation in the absence of CD98hc or expression of C330S, including proteasomal or

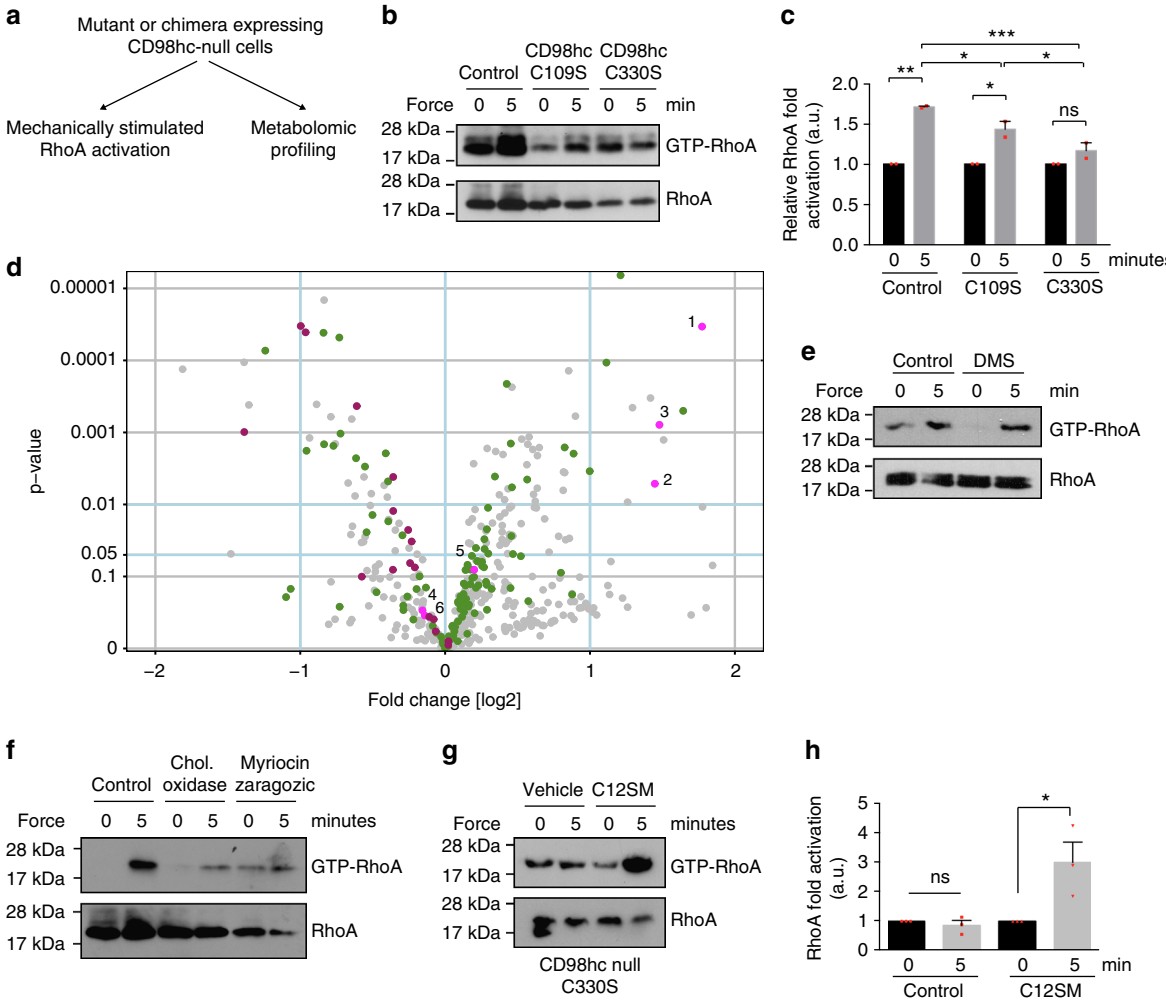

**Fig. 4** CD98hc regulates sphingolipid biosynthesis and integrin mechanosensing. **a** Schematic strategy to identify CD98hc mutants regulating simultaneously integrin mechanosensing and cell metabolism. **b** Re-expression of C109S CD98hc mutant in CD98hc-null cells but not of C330S mutant rescues mechanically coupled RhoA activation, one experiment representative of $n = 2$. **c** Quantification of RhoA activation in control, C109S or C330S-expressing CD98hc-null cells. Means are plotted with s.e.m. as error bars from $n = 2$ experiments. $^{*}P < 0.05$, $^{**}P < 0.01$, $^{***}P < 0.001$ in a two-way ANOVA. **d** Volcano plot depicting metabolites fold changes versus $p$ value in C330S versus CD98hc re-expressing CD98hc-null cells. Pink, long chain based sphingoids, ceramides and phytoceramides; magenta, glycosphingolipids and sphingomyelins; green, amino acid metabolism. Numbered metabolites are analogous to the numbering in Fig. 5i. 1, 3-ketosphinganine; 2, sphinganine; 3, N-palmitoyl-sphinganine; 4, N-palmitoyl-sphingosine; 5, palmitoyl dihydrosphingomyelin; 6, sphingosine. $P$ values were calculated using a Student's $t$ test, values are reported in Supplementary Data 1. **e** Inhibition of S1P synthesis with dimethyl-sphingosine (DMS) treatment at 10 μM for 1 h does not impair mechanically coupled RhoA activation triggered by FN-coated paramagnetic beads, one experiment representative of $n = 2$. **f** Sphingolipid and cholesterol depletion impairs mechanically coupled RhoA activation, one experiment representative of $n = 2$. **g**, **h** Exogenous supply of C12 sphingomyelin to C330S-expressing cells restores mechanically coupled RhoA activation. H is representative of $n = 3$ experiments. Means are plotted with s.e.m. as error bars in I from $n = 3$ experiments. $^{*}P < 0.05$ in a Student's $t$ test

lysosomal degradation. However, DES2 proper folding requires intervention of molecular chaperones since inhibition of Hsp90 family chaperones resulted in downregulation of the amount of DES2 (Fig. 5j, k). CD98hc has previously been reported to facilitate the expression of instable transmembrane proteins such as MCTs, Glut1, and LAT1[29]. Therefore, we envisioned that, somehow, CD98hc may be required for the Hsp-dependent folding and stabilization of DES2. We found, by coimmunoprecipitation, that DES2 associated constitutively with grp94, Hsp90 counterpart in the endoplasmic reticulum (ER), in CD98hc-null cells as well as in C330S cells but not in control cells (Fig. 5l). However, upon inhibition of grp94 with geldanamycin, we observed increased sequestration of grp94 by DES2 only in control cells. This indicates that loss of CD98hc prevents proper folding of DES2 and triggers intervention of molecular chaperones that can be mimicked by use

of geldanamycin. We also observed that grp78/Bip1, Hsp70 counterpart in the ER, was recruited to the complex in CD98hc-null and C330S cells but absent upon geldanamycin treatment. In order to link DES2 and mechanosensing, we assessed the consequences of DES2 silencing on mechanosensing. DES2 depletion impaired mechanically coupled RhoA, SFK, and GEF-H1 activation, and promoted sequestration of GEF-H1 in membranes (Fig. 6a–d) without altering the integrin repertoire present at the surface of the cells (Supplementary Fig. 6). Also, it prevented mechanical activation of YAP/Taz similarly to loss of CD98hc (Fig. 6e). Surprisingly, TFM measurements showed that DES2 depleted cells generated more forces than control cells (Fig. 6f) indicating that while DES2 is clearly essential to regulate RhoA activation, its contribution to other multifactorial mechanically related processes may be more complex.

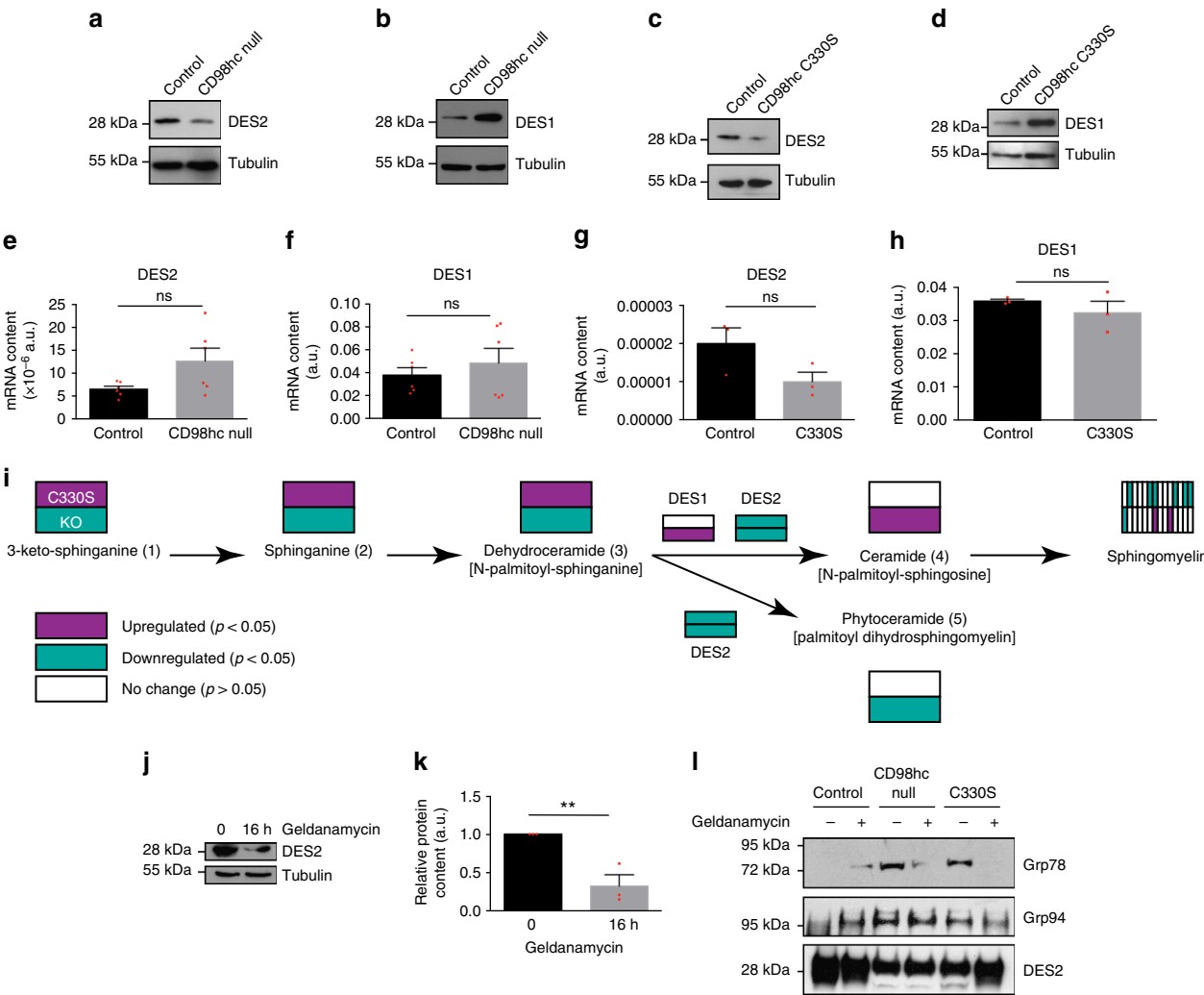

**Fig. 5** CD98hc determines the level of DES2 in fibroblasts. **a–d** Loss of CD98hc or expression of C330S mutant reduce the expression of DES2 and induces DES1. One experiment representative of at least $n = 3$ experiments. **e–h** Levels of DES1 and DES2 mRNA as measured by RT-qPCR on total mRNA from CD98hc-null cells or C330S reconstituted null cells. Means are plotted with s.e.m. as error bars from, respectively, $n = 6$ or $n = 3$ experiments. Differences were not statistically different in a Student's $t$ test. **i** Schematic representation of the de novo sphingolipid biosynthesis pathway presenting variations of metabolites. Numbers refer to metabolites depicted and numbered in the volcano plot of Fig. 4d. Metabolites and $p$ values are reported in Supplementary Data 1. $P$ values were calculated using a Student's $t$ test. DES1/2 variations were extracted from results presented in Fig. 5a–d and Supplementary Figure 5A–D. **j, k** Hsp90 inhibition in control cells reduces the level of DES2 expressed in cells. Right panel, means are plotted with s.e.m. as error bars from $n = 3$ experiments, $^{**}P < 0.01$ in a Student's $t$ test. **l** Grp78 and Grp94 co-immunoprecipitate with DES2 upon loss of CD98hc or geldanamycin treatment. Control, CD98hc-null, or C330S-expressing cells were treated with geldanamycin for 16 h prior to cell lysis and DES2 immunoprecipitation. Precipitates were loaded on SDS–PAGE and analyzed by Western blot. One experiment representative of $n = 2$

**Sphingolipids regulate Src and GEF-H1 mechanical activation.**
While our results establish that CD98hc connects sphingolipid biosynthesis and integrin mechanosensing, the molecular basis behind this cross-regulatory mechanism remained unclear. Therefore, we dissected the regulatory pathways linking force application on integrins to RhoA activation[19]. First, we monitored the activation of upstream regulators of RhoA in this pathway by GST-RhoA 17A pull-down assay[30]. Upon loss of CD98hc, both RhoGEFs LARG and GEF-H1 were unresponsive to force application on integrins (Fig. 7a and Supplementary Fig. 7a). Upstream of these GEFs, ERK1/2 were activated by force application while SFK remained unresponsive (Fig. 7b), indicating that impaired RhoA activation resulted from defective GEF-H1 and Src kinases activation. Considering the regulation of sphingolipids by CD98hc, we directed our focus to the regulation of these two proteins at the level of cellular membranes. As counterintuitive as it may be, we found that GEF-H1 was

constitutively recruited at membranes in CD98hc-null cells, regardless of force application (Fig. 7c), much like in CD98hc-deprived HeLa cells (Supplementary Fig. 7b). Similarly, in C330S-expressing CD98hc-null fibroblasts, GEF-H1 was constitutively recruited at membranes (Supplementary Fig. 7c). Then, we precisely dissected the activation of GEF-H1 coupled to its membrane localization by cell fractionation and GST-RhoA 17A pull-down. In control cells, mechanical stimulation induced both catalytic activation and membrane recruitment of GEF-H1. In CD98hc-null cells, GEF-H1 was constitutively recruited to membranes, as previously observed. The amount of active GEF-H1 in membranes was similar to that of control cells indicating that the bulk of GEF-H1 recruited to membranes in CD98hc-null cells must be catalytically inactive. Upon mechanical stimulation, part of the cytosolic GEF-H1 could be catalytically activated without being significantly translocated to the membrane as compared to control cells indicating that enzymatic activation

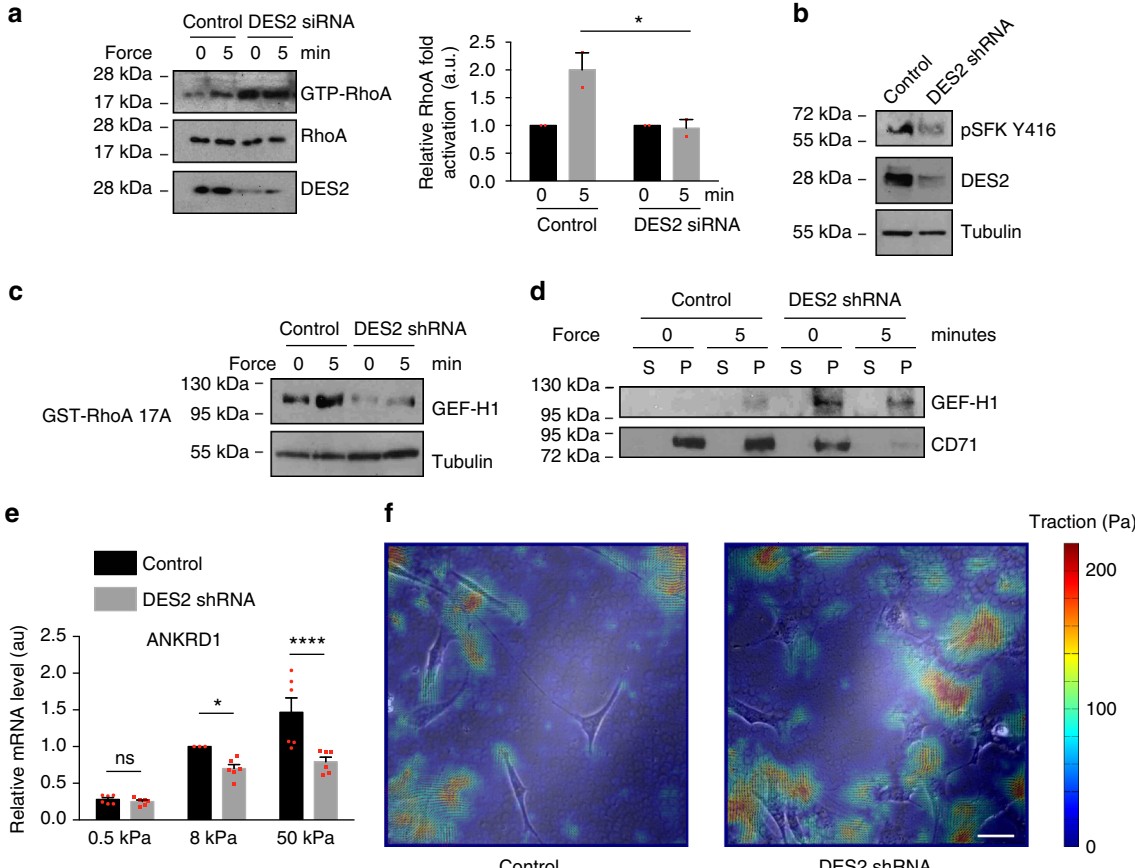

**Fig. 6** Depletion of DES2 recapitulates most of defects observed upon depletion of CD98hc. **a** Depletion of DES2 by siRNA impairs RhoA activation by mechanical force application on integrins. Right panel, quantification of RhoA activation, means are plotted with s.e.m. as error bars from $n = 2$ experiments, $^*P < 0.05$ in a two-way ANOVA. **b** Depletion of DES2 by shRNA impairs SFK phosphorylation at Y416. One experiment representative of $n = 2$. **c** Depletion of DES2 impairs mechanically induced GEF-H1 activation. One experiment representative of $n = 2$. **d** Depletion of DES2 triggers GEF-H1 sequestration in cell membrane. s supernatant, p membrane pellet. One experiment representative of $n = 2$. **e** DES2 depletion partially inhibits mechanically coupled YAP/Taz activation as measured by RT-qPCR of ANKRD1 transcript. Means are plotted with s.e.m. as error bars from $n = 6$ with $^*P < 0.05$ and $^{****}P < 0.0001$ in a two-way ANOVA. **f** Depletion of DES2 impairs generation of traction forces. Traction force microscopy was performed on subconfluent cells. Scale bar is 50 μm

coupled to membrane recruitment of GEF-H1 were both required for further activation of RhoA (Fig. 7d). This highlights the recruitment of catalytically active GEF-H1 to membranes as a critical step and prompted us to consider the contribution of sphingolipids and sphingolipid-containing structures such as CEMM in this process, either directly on GEF-H1 or indirectly via other intermediates. Regarding these potential intermediates, integrins are notably interesting since they participate to the recruitment of GEF-H1 in adhesions[31], their dynamics is affected by modifications of membrane composition[32] and their role is pivotal in mechanosensing. We found that, although integrin cell surface expression and activation were not affected by loss of CD98hc (Supplementary Fig. 1f), their membrane dynamics and recycling were altered. As measured by fluorescence recovery after photobleaching (FRAP), we found that rates of diffusion of integrins α5 and β3 were not statistically significantly altered in adhesions upon loss of CD98hc (Fig. 7e, f, and Supplementary Fig. 8a–d). However, both integrins mobile fractions in adhesions were significantly reduced upon loss of CD98hc or expression of C330S mutant as compared to control cells (Fig. 7g, h). Integrin recycling was also altered upon loss of CD98hc or expression of C330S. Integrin β1 was recycled at a slower rate in CD98hc-null cells as compared to control cells as they were already targeted back to the membrane within an hour in control cells (Fig. 7i). An

intermediate recycling rate was observed in C330S cells (Supplementary Fig. 8e, f). This also indicates that integrins were less mobile at the plasma membrane because of decreased sphingolipid content, which could participate to the sequestration of GEF-H1 at the membrane. As indicated, we also hypothesized that sphingolipid-rich domains such as CEMM may regulate GEF-H1 localization. Indeed, regulation of several Src family members including Yes and Fyn also relies on recruitment in CEMM[33], which is interesting since Fyn regulates RhoA activation downstream of mechanical force application on integrins[19]. Therefore, we suspected that Src family kinases and GEF-H1 may shuttle through CEMM, by direct or indirect interaction, during their cytosol-membrane cycling. Hence, we monitored their recruitment in CEMM by membrane fractionation and sucrose gradient purification. Both Src kinases and GEF-H1 were present in CEMM of control cells while their levels were severely reduced in CD98hc-null cells (Fig. 8a, b). Notably, starting from similar amounts of biological material, flotillin levels were decreased indicating that CEMM integrity was most likely affected by loss of CD98hc. Additionally, immunofluorescence staining of cholesterol and GEF-H1 showed that they colocalized in membrane protrusions only in control cells and that GEF-H1 is sequestered in the plasma membrane of CD98hc-null and DES2 shRNA cells (Fig. 8c and Supplementary Fig. 9a, b). While both Src kinases

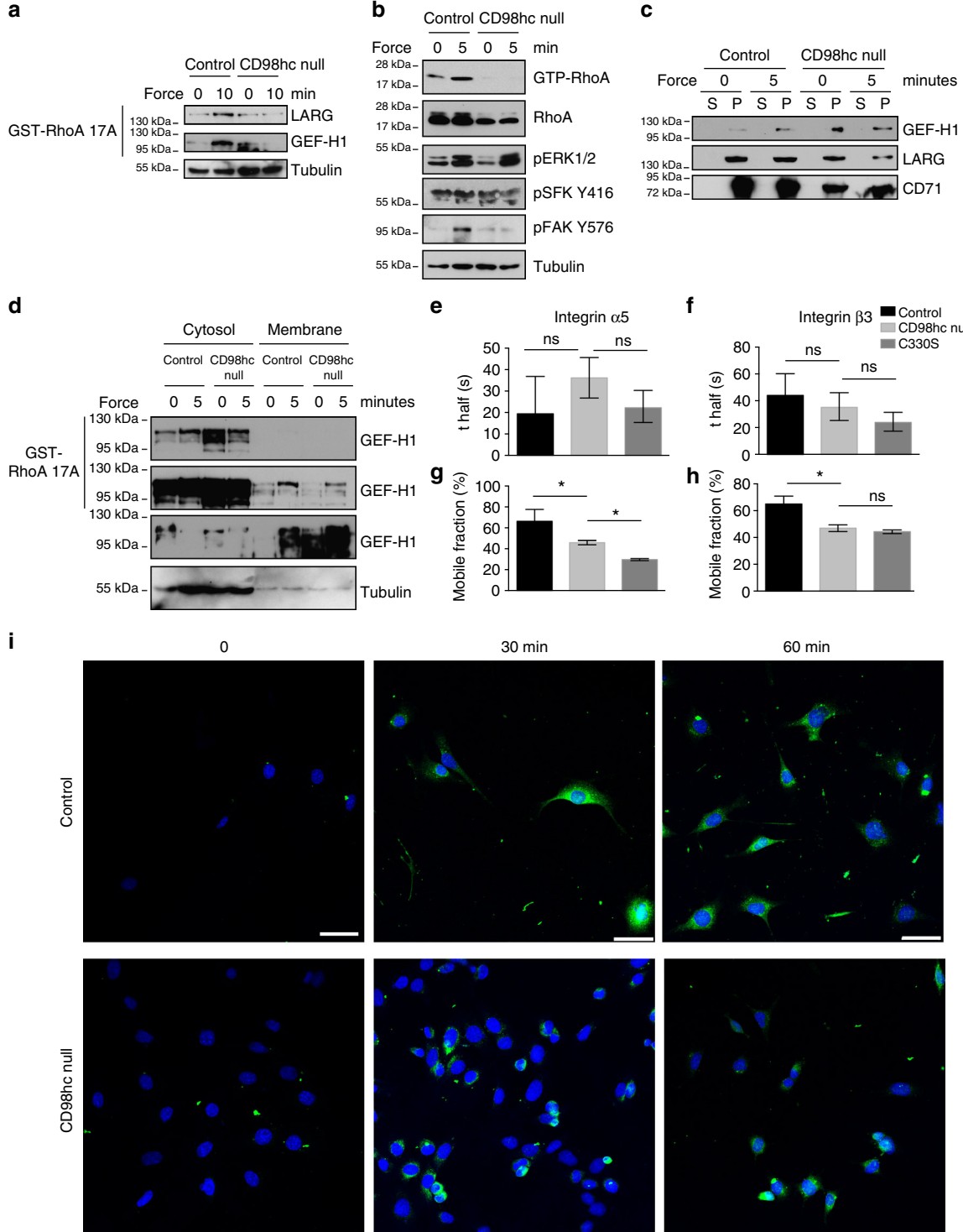

**Fig. 7** Loss of CD98hc impairs GEF-H1and Src family kinase activation, and integrin dynamics. **a** Loss of CD98hc impairs LARG and GEF-H1 activation by mechanical force application on integrins. One experiment representative of $n = 2$. **b** Loss of CD98hc impairs Src activity but does not affect ERK1/2 phosphorylation by mechanical force application on integrins. Src activity was monitored by tracking substrate FAK Y576 phosphorylation and SFK phosphorylation at Y416. **c** Loss of CD98hc induces constitutive recruitment of GEF-H1 to the membrane. s supernatant, p membrane pellet. One experiment representative of $n = 3$. **d** Control or CD98hc-null dermal fibroblasts were stimulated with FN-coated magnetic beads. Cells were lysed and lysates were fractionated into cytosolic (s) and membrane (p) fractions by centrifugation. Active RhoGEFs was pulled-down from each fraction with GST-RhoA17A beads. One experiment representative of $n = 2$. **e**, **f** Diffusion rate (*t*-half) of integrin α5 and β3 as calculated from the curve fit generated from the FRAP measurements performed on $n = 27$, 41, and 67; and 73, 36, and 57 adhesions, respectively, for control, CD98hc-null and C330S cells on integrin α5 and β3. Error bars are 95% CI calculated from that fit. **g**, **h** Integrin α5 and β3 mobile fraction as calculated from the curve fit generated from the FRAP measurements. Error bars are 95 CI calculated from that fit. **i** Trafficking of integrin β1 in control or CD98hc-null cells. Integrin β1 was labeled with Alexa 488 coupled antibody then integrin trafficking was chased for indicated time. Extracellular staining was quenched and only intracellular labeled integrin is observed. Scale bar is 50 μm

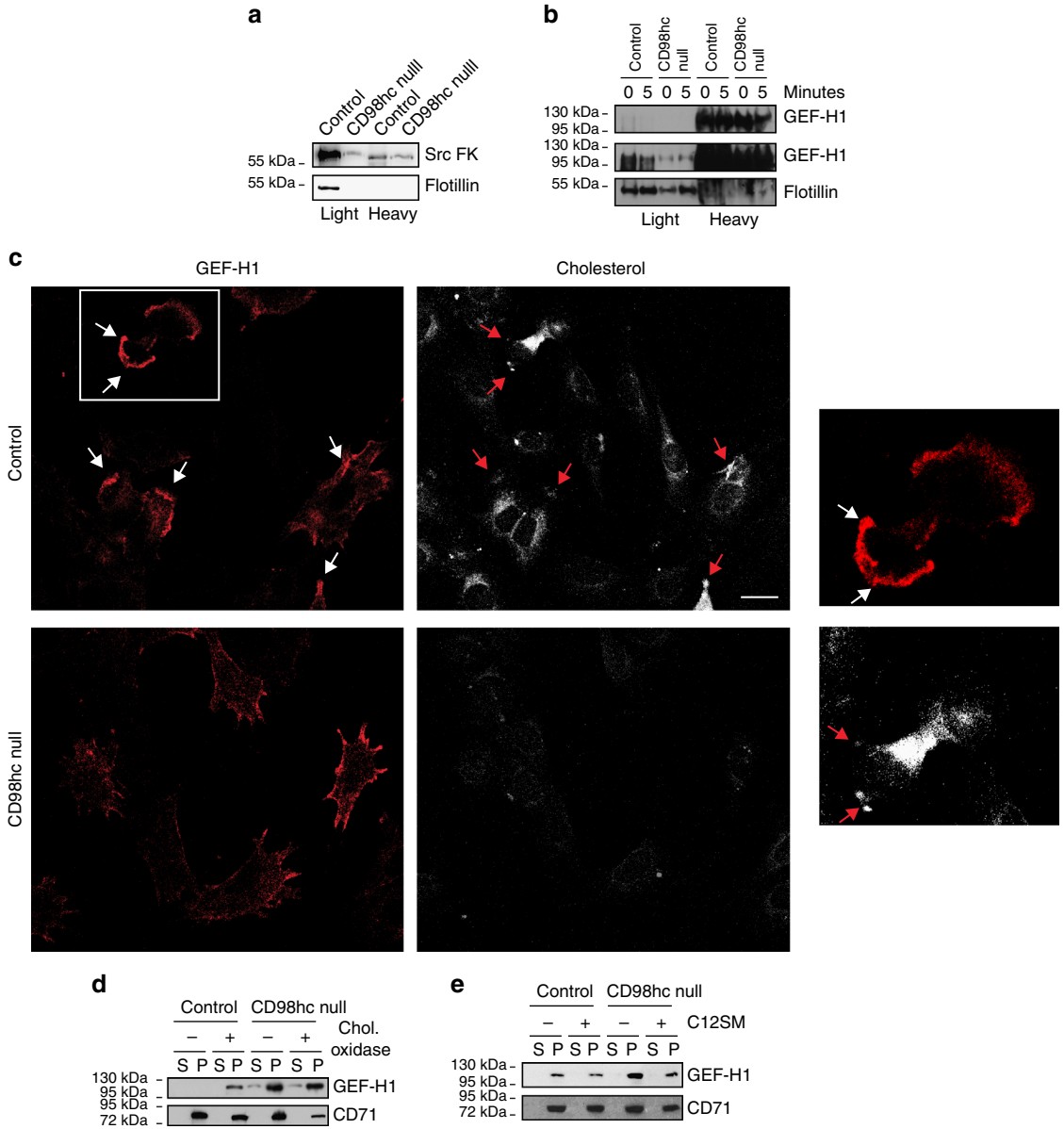

**Fig. 8** Loss of CD98hc affects GEF-H1 recruitment in membrane microdomains. **a**, **b** Loss of CD98hc impairs recruitment of src kinases (**a**) and GEF-H1 (**b**) in CEMM. One experiment representative of *n* = 2. **c** GEF-H1 and cholesterol colocalize in control cells (arrows). Endogenous GEF-H1 and cholesterol were stained using an anti-GEF-H1 antibody or filipin III respectively. Right panel is a magnified crop of the white squared box. Scale bar is 50 μm. **d** Cholesterol depletion induces accumulation of GEF-H1 in membranes. **e** C12SM exogenous supply induces a decrease in the amount of GEF-H1 constitutively recruited to membrane fractions in CD98hc-null cells to levels similar to that of control cells

regulation by CEMM[33] and GEF-H1 regulation downstream of force application on integrins have been characterized[19,31], virtually nothing is known about the regulation of GEF-H1 cycling between cytosol and membrane by sphingolipids and cholesterol. We speculated that GEF-H1 may have to shuttle through membrane microdomains during its cytosol-membrane cycling in order to be released from the membrane. This would explain the counterintuitive accumulation of GEF-H1 in membranes of CD98hc-null cells. Indeed, cholesterol depletion induced accumulation of GEF-H1 in membranes of control cells much like that of untreated CD98hc-null cells (Fig. 8d) indicating that cholesterol is required for GEF-H1 to be released from the membrane. Conversely, exogenous supply of sphingomyelin in CD98hc-null cells prevented accumulation of GEF-H1 in membrane fraction resulting in GEF-H1 levels similar to that of control cells in these fractions (Fig. 8e). In conclusion, we show

that loss of CD98hc induces disruption of membrane microdomains because of decreased availability of sphingolipids and cholesterol which results in improper spatiotemporal regulation of Src kinases and GEF-H1. This defective regulation prevents proper activation of RhoA upon mechanical stimulation and adequate cellular response.

## Discussion

In the present study, we characterize a novel function for CD98hc as a regulator of integrin mechanosensing but foremost, we uncover an unanticipated cross-regulatory mechanism between cell metabolism and integrin mechanosensing regulated by the integrin coreceptor and amino acid transporter CD98hc (SLC3A2). For years, the consensual role of CD98hc has been that of a bifunctional protein independently regulating two separate

cellular functions. Originally, CD98hc had been characterized as a regulatory subunit associating with different SLC7 light chains to constitute amino acid transporters of the HAT family[8]. It later emerged as a regulator of integrin function[9] and several recent lines of evidence seem to indicate that CD98hc may regulate integrin mechanosensing[18]. However, despite this physical link, the question as to whether these two functions could cross-regulate each other has never been addressed. Here, for the first time, we describe at the molecular level how CD98hc regulates integrin-mediated mechanosensing by affecting RhoA activation. We show this regulation has biological consequences at the cellular level as well as at the tissue level since invalidation of CD98hc in the dermis results in aberrant skin mechanical homeostasis much like what happens during aging. This is not completely surprising since CD98hc expression decreases during aging in skin[13]. Overall, these defects resulted from defective mechanical sensing which affected several cellular responses including mechanical regulation of YAP/Taz activity. Our findings parallel those made in the medaka fish *Oryzias latipes*, showing that loss of YAP function in vivo induces loss of mechanical tension in the tissue[34]. Although novel, these findings are not particularly surprising since several integrin associated proteins also regulate integrin mechanosensing and mechanosignaling including talin, p130Cas and vinculin[35–37].

However, the peculiarity of this regulation resides in the fact that it occurs indirectly, via cell metabolism thereby generating a cross-regulation between cell metabolism and integrin mechanosensing. By screening several CD98hc mutants and chimeras, we identified the C330S mutant as being simultaneously defective in integrin mechanosensing and de novo sphingolipid biosynthesis. This differed quite from what we expected since amino acid metabolism was unaffected, indicating a new function for CD98hc independent from amino acid transport. Detailed analysis of this metabolic pathway indicated that loss of CD98hc or expression of C330S mutant would impair the conversion of dihydroceramide into ceramide, an enzymatic step catalyzed by delta-4-desaturases of the DES family. We could link that enzymatic defect to a downregulation of the most prevalent enzyme in the skin, DES2. While it is still unclear how CD98hc precisely regulates the expression of this enzyme several lines of evidence led us to envision that CD98hc may contribute to its folding via molecular chaperones of the Hsp family.

This downregulation of DES2 blocks the de novo sphingolipid biosynthesis pathway. Sphingolipids are essential membrane constituents that participate to the regulation of transmembrane proteins dynamics and to the assembly of specific membrane microdomains, originally named lipid rafts. Equally important, sphingolipids are bioactive lipids that behave as second messengers. Although, we cannot formally exclude GEF-H1 being regulated by such second messenger sphingolipids, we rather hypothesized a regulation via alteration of membrane composition and membrane microdomains since sphingolipid and cholesterol content are both affected by loss of CD98hc. Indeed, CD98hc association to its light chain LAT1 is cholesterol dependent[38]. We found that sphingolipid downregulation, upon loss of CD98hc, altered integrin behavior which, in light of their role on GEF-H1 recruitment, may participate to this regulation. Additionally, this decrease in sphingomyelin availability affects the behavior of membrane microdomains and the regulation of the proteins transiting through. We show this is a crucial element explaining how Src and GEF-H1 are regulated by CD98hc to activate RhoA downstream of force sensing by integrins. Src kinases have long been described as regulated by cholesterol-enriched membrane microdomains[33], for GEF-H1, this constitutes an important advance in the understanding of its regulation during mechanosensing. GEF-H1 is a Rho-specific GEF which associates with microtubules in its inactive

state and is generally released whenever activated[39,40]. GEF-H1 has been linked to mechanosignaling in response to various types of mechanical stimulations[19,41]. In the particular case of integrin mechanosensing, GEF-H1 is recruited by αvβ3 to the adhesion complex where it is activated by ERK1/2 phosphorylation downstream of force application on the α5β1 integrin receptor[19,31]. Here, we refine this mechanism by showing that proper activation of GEF-H1 requires sphingolipids and shuttling through membrane microdomains. Indeed, previous studies had already linked GEF-H1 to membrane microdomains during mechanosensing since it has been shown to colocalize with flotillin2, a protein residing in CEMM, in the uropod of neutrophils upon shear stress[41]. Overall, these novel findings can be integrated in the current knowledge about GEF-H1 regulation during mechanosignaling as one may envision that GEF-H1 is indeed recruited to the adhesion complex and the membrane by αvβ3, but needs to shuttle through CEMM to be released into the cytosol.

Finally, the major finding and conceptual advance of our work is that integrin mechanosensing can be regulated by cell metabolism. Whereas others have reported the HMG-CoA reductase pathway affecting mechanotransduction via the prenylation of RhoA[42], our discovery that sphingolipid metabolism regulates integrin mechanosignaling suggests a direct and novel level of cross talk between the metabolic state of cells and their response to mechanical forces. Sphingolipid synthesis starts with the condensation of serine and palmitoyl-coA into 3-ketosphinganine meaning that alterations in the availability of either of these two metabolites will affect the ability of cells to generate sphingolipids. This can occur if fatty acids are used for other purposes such as generating energy through fatty acid β-oxidation for instance. More generally, this may occur under conditions where cell metabolism is altered such as in the case of metabolic disorders or cancer. This one is particularly interesting since both altered mechanosensing, environmental mechanical properties, and cell metabolism are hallmarks of cancer and tumor progression[2,43,44]. Therefore, modifications of cell metabolism occurring during tumor progression, may contribute to the alteration of tumor mechanosensing. Interestingly, a recent report indicates that cross-regulation from mechanosensitive adhesion complexes toward cell metabolism may also occur[17]. This clearly supports and strengthens our own findings and suggests that this type of cross-regulation may be reciprocal and generalized. Together, these cross-regulatory mechanisms emerge as a novel general regulatory framework integrating these two fundamental cellular processes.

## Methods

**Reagents**. Unless mentioned all chemicals were from Sigma-Aldrich. Unless mentioned all tissue culture media were from Thermo Scientific Life. Polyacrylamide hydrogels were from Matrigen. CNF1 was from Cytoskeleton. C12 SM was from Avanti Lipids.

**Antibodies**. The mouse monoclonal anti-RhoA 26C4 (WB 1/500), anti-CD71 H68.4 (1/1000) and anti-YAP 63.7 (1/1000) antibodies, the rabbit polyclonal anti-LARG H-70 (1/500), anti-LOX1 (1/500) and anti-CD98hc H-300 (1/1000) antibodies were from Santa Cruz Biotechnology. The rabbit monoclonal anti-GEF-H1 55B6 (1/500), anti-Src 36D10 (1/500), anti-Grp94 D6X2Q (1/1000), anti-Grp78/BiP C50B12 (1/1000) and anti-Vimentin D21H3 (1/500) antibodies, and the rabbit polyclonal anti-phospho Y416 Src (1/500) and anti-flotillin-1 (1/1000) (#3253) antibodies were from Cell Signaling Technology. The mouse monoclonal anti-Tubulin TUB2.1 (1/5000) and anti-diphosphorylated ERK1&2 M8159 (1/10,000) antibodies were from Sigma-Aldrich. The rabbit polyclonal anti-DES2 (PA5-24082) (1/1000), anti-DES1 (PA5-42741) (1/1000) and anti-pY576 FAK (44652 G) (1/500) antibodies were from Thermo Fischer Life. The mouse monoclonal anti-CD98hc RL388 antibody was from eBioscience. The mouse monoclonal anti-Rac1 23A8 (1/1000) antibody was from Millipore. The function blocking mouse monoclonal anti human integrin β3 antibody LM609 was a gift of Dr. Laetitia Seguin (IRCAN, Nice, France). The hamster monoclonal anti-mouse CD61/integrin β3 2C9.G2, anti-mouse CD49e/integrin α5 HMa5-1, anti-mouse CD49a/

integrin α1 Ha31/8 were from BD Biosciences (all used at 1/100 dilution by FACS). The mouse monoclonal anti-GEF-H1 (B4/7) used for immunofluorescence (1/500) was from Abcam (Cambridge, UK). The rat monoclonal anti-integrin β1 clone MB1.2 (IF 1/1000) was from Merck Millipore (Temecula, CA, USA).

**Mice**. All procedures were approved by the Institutional Animal Care and Use Committee at the University of Nice-Sophia Antipolis, Nice, FR (CIEPAL-NCE191). CD98hc conditional null mice, $CD98hc^{fl/fl7}$ were crossed with $Fsp1$-$Cre$ transgenic mice #012641 from The Jackson Laboratory (Bar Harbor, ME, USA). For all in vivo experiments, $[Fsp1$-$Cre,CD98hc^{fl/fl}]$ mixed-background mice and age matched (3 months old) $CD98hc^{fl/fl}$ littermate controls were used. Mice were allocated to experimental groups on the basis of their genotype and age, regardless of sex. No randomization of animals was performed. All quantifications were performed blinded as the identity of the sample was concealed from the staff performing the experiment. Monte-Carlo simulation was used to determine group size.

**EM measurements**. The mechanical properties of the samples were studied using a BioScope Catalyst atomic force microscope (Bruker) coupled with an optical microscope (Leica DMI6000B).

For each sample, 5–8 zones of 140 μm² were measured by "Point and Shoot" and at least 200 force–distance curves spaced from each other by, at least, 20 μm, were collected. The experiments of nanoindentation were performed using either a probe with a Borosilicate Glass spherical tip (5 μm of diameter, Novascan) or a probe with silicon pyramidal tip (probe SNL-D, side angle 23°, Bruker). Both probes have a V-shaped cantilever with a nominal spring constant of 0.06 N/m. Indentations were performed with a velocity of 6.51 μm/s, in relative trigger mode and by setting the trigger threshold to 5 nN. The apparent Young's modulus was calculated using the NanoScope Analysis 1.50 software (Bruker), fitting the force curves either to the Hertz or to the Sneddon indentation model, depending on the probe used in the experiment (spherical or pyramidal, respectively), and using a Poisson's ratio of 0.5. All force–distant curves having not a clear base line, a maximum above 5 nN or a change of slope in the region of the fitting (minimum and maximum force fit boundary 5% and 25%, respectively) were discarded. The apparent Young's modulus distribution was fitted to a nonparametric kernel function to extract the most probable value of $E$[45]. E values were obtained from at least 60 measurements on skin samples from $n = 5$ and 4 mice, respectively in each condition.

**Collagen and elastin staining**. Picrosirius red analysis was achieved through the use of paraffin sections of back skin stained with 0.1% picrosirius red (Direct Red80, Sigma) and counterstained with Weigert's iron haematoxylin (#CO231, Diapath, Martinengo, Italy) to reveal fibrillar collagen. Sections were then analyzed using a zeiss microscope fitted with an analyzer and a polarizer oriented parallel and orthogonal to each other and quantified using ImageJ software. Elastin staining was performed using Accustain Elastin stain from sigma-Aldrich (HT25A-1KT) according to manufacturer's instructions. Length of fibers was quantified using imageJ.

**qPCR/RNA preparation**. RNAs were extracted from back skin samples using Trizol reagent (GIBCO). Reverse transcription was performed on total RNA using Superscript II reverse transcriptase (Invitrogen) according to manufacturer's instructions. Sets of specific primers were used for amplification using 7900HT Real Time PCR System (AppliedBiosystems, Foster USA). Samples were normalized to rplp0 using the ΔCt method. Statistical significance was determined with Student's $t$ test or two-way ANOVA for multiple comparisons as stated in the Figure legend.

**qPCR/siRNA oligos**. SiGENOME siRNA SmartPools targeting mouse YAP1 and wwtr1 were from Dharmacon. Primers used for qPCR: ANKRD1 F 5′ACGCA GACGGGAACGGAAGC3′, R 5′ TGCGGCACTCCTGACGTTGC3′; CTGF F5′ GCGAGCCAACTGCCTGGTCC3′, R5′CAGCTTCGCAGGGCCTGACC3′; LOX F5′CGTCCACGTACGTGCAGAAG3′, R5′CCTGTATGCTGTACTGGCCA GAC3′; LOXL1 F5′CCGCAGCAGTTCCCCTATC3′, R5′CGCGGGATCG TAGTTCTCA3′; TG2 F5′AGTATGAGCATGGGCAACGA3′, R5′ATACAG GGGATCGGAAAGTG3′; mDES1F: TCT TGA AGG GAC ACG AAA C; mDES1R: CCG TCA CAA AGT CAT AGA GC; mDES2F:ATA TGT TCC TGA AGG GC; mDES2 R: TGC GCT AAA CCC TGG AGT AG.

**Cell culture**. Primary mouse dermal fibroblasts were isolated from $[Fsp1$-$Cre$-$CD98hc^{fl/fl}]$ adult skin using enzymatic digestion. Briefly, skin was incubated on dispase for 30 min at 37 °C, then in collagenase I and IV to dissociate the dermal cellular contents. Fibroblasts were cultured in DMEM-H (Invitrogen) culture medium containing 10% (v/v) FBS (HyClone), 20 mM HEPES, pH 7.3 (Invitrogen), 0.1 mM nonessential amino acid (Invitrogen), 0.1 mM β-mercaptoethanol (Sigma-Aldrich), and 2 mM L-Glutamine (Invitrogen) at 37 °C and 5% CO₂. HeLa cells were from ATCC (ATCC CCL-2) and were grown in DMEM-H containing 10% FBS and 2 mM L-Glutamine at 37 °C and 5% CO₂. HeLa cells were used on the

basis of their epithelial nature to show that this regulatory mechanism was not restricted to fibroblasts but can extend to other cell types.

Primary Dermal fibroblasts have been immortalized by transfection with SV40. All cell cultures were tested negative for mycoplasma contamination by PCR.

**Western blotting**. Whole cell lysates were prepared using 50 mM Hepes pH 7.4, 250 mM NaCl, 5 mM MgCl2, 1% triton X-100, 0.1% sodium dodecyl sulfate (SDS), 5 mM dithiothreitol (DTT) and mini EDTA-free protease inhibitors (Roche) at 4 °C. Protein lysates were quantified using the bicinchoninic acid method (Thermo Scientific Pierce), loaded on SDS–PAGE, and analyzed by Western blotting.

**Subcellular fractionation**. Cells were washed and incubated for 10 min with ice-cold hypotonic lysis buffer (10 mM HEPES at pH 7.3, 1.5 mM MgCl₂, 5 mM KCl, 1 mM DTT and protease inhibitors). Cells were scraped and homogenized with 20 strokes of a Dounce homogenizer. Homogenates were centrifuged at 700$g$ for 3 min to pellet nuclei and intact cells. The supernatants were spun at 40,000$g$ for 30 min at 4 °C and the pellets were gently washed once with hypotonic lysis buffer.

**Sucrose gradient purification**. Cells were lysed in ice-cold lysis buffer (150 mM Na2CO3 pH 11, 2 mM EDTA and protease inhibitors). Cells were scraped and homogenized with 15 passes through a 25 G needle. Lysates were mixed with 80% w/v sucrose to reach 40% sucrose concentration. A 5, 35, and 40% discontinuous sucrose gradient was set up in 11 mL ultracentrifuge tubes with lysates at the bottom in the 40% sucrose fraction. Gradients were centrifuged for 20 h at 247,000$g$ (38,000 rpm) at 4 °C in a swinging bucket SW41Ti rotor. One milliliter fractions were collected from the top of the gradient. Fractions were concentrated and analyzed by Western blot.

**GST–RBD/PBD pull-down and G-LISA assay**. Activation of RhoA or Rac1, respectively, was measured in a GST–RBD or GST–PBD pull-down assay as described previously. In brief, cells were lysed for 10 min in 25 mM HEPES pH 7.3, 150 mM NaCl, 5 mM MgCl₂, 0.5% Triton X-100, 0.1% SDS, 10 mM NaF, 5 mM dithiothreitol (DTT), and protease inhibitors at 4 °C. Triton-X-100-insoluble material was removed by centrifugation for 10 min at 9500$g$ and the lysates were incubated for 40 min with 50 μg of immobilized GST–RBD or GST–PBD at 4 °C, to measure RhoA or Rac1/Cdc42 activity, respectively. G-LISA assay (Cytoskeleton, CO, USA) was performed according to manufacturer's protocol.

**Immunoprecipitation**. Cells were grown for 24 h prior to treatment with geldanamycin for 16 h. Cells were lysed in 25 mM HEPES pH 7.3, 150 mM NaCl, 5 mM MgCl₂, 0.5% Triton X-100 and protease inhibitors at 4 °C. Triton-X-100-insoluble material was removed by centrifugation for 10 min at 9500$g$. Immunoprecipitation was performed using 5 μg of polyclonal DES2 antibody overnight at 4 °C. Lysates were then incubated with 10 μL of protein A-coupled dynabeads for 1 h, precipitated and washed 3 times using a permanent magnet. Immunoprecipitates were analyzed by Western blot.

**Immunofluorescence**. Cells were fixed in PBS 100 mM sucrose for 30 min, permeabilized with PBS 0.5% Triton X-100 for 3 min before incubation with anti-YAP antibody 1/100 in PBS for 2 h at room temperature. Cells were washed three times and incubated with fluorescently-labeled phalloidin PBS supplemented with DAPI for 1 h. Cells were washed three times before incubation with fluorescently labeled secondary antibody 1/250 in PBS for 1 h. Hydrogels were mounted in mowiol and imaged on a Zeiss LSM510 confocal microscope.

Cholesterol staining was performed according to manufacturer's protocol using the Cholesterol Cell-Based Detection Assay Kit (10009779) from Cayman Chemicals (Ann Arbor, Michigan, USA). Cells were permeabilized with PBS 0.5% Triton X-100 for 3 min before incubation with anti-GEF-H1 antibody (B4/7) for 1 h. Cells were washed three times before incubation with fluorescently labeled secondary antibody 1/250 in PBS for 1 h. Coverslips were mounted in mowiol and imaged on a Zeiss LSM510 confocal microscope.

**Integrin FRAP**. Cells were transfected with GFP-integrin β3 or GFP-integrin α5 chimeras using Fugene HD (Promega). Cells were grown for 72 h, split and seeded on FN-coated coverslips for 12 h. FRAP experiment was performed on an inverted Zeiss LSM 880 confocal, at 37 °C and 5% CO₂. FRAP was performed on fluorescent focal adhesions by bleaching fluorescence with 10 iterations of illumination at 100% of laser's power after acquisition of two images to evaluate basal, pre-bleaching level of fluorescence. For each acquisition, background fluorescence was evaluated by measuring fluorescence in a dark irrelevant area. Similarly, fluorescence was also measured on an unbleached focal adhesion to control and correct for laser power fluctuation and overall bleaching of fluorophore during measurement. Fluorescence recovery was measured by acquiring images every five seconds for as long as recovery was occurring. Data normalization and analysis was performed using easyFRAP-web at https://easyfrap.vmnet.upatras.gr. Normalized data was fitted to an exponential equation $y = k - ae^{-bx} - ce^{-dx}$ using Matlab in order to evaluate the rate of diffusion ($t$-half) and the mobile fraction. Fitness of the fit and other statistical information are reported in the legend of the Figure.

**Integrin trafficking**. The anti-integrin antibody, clone MB1.2, was labeled with Alexa 488 using the APEX Alexa Fluor 488 antibody labeling kit (Thermo) according to the supplier's protocol. Cells were grown on FN-coated coverslips for 12 h prior to the assay. Cells were incubated in complete culture medium supplemented with the Alexa 488 labeled anti-integrin antibody (1/1000) for 20 min at 4 °C. Cells were washed twice with complete medium at 4 °C then incubated at 37 °C in complete medium for the indicated times. Cells were fixed in 3.7% paraformaldehyde for 30 min. Cells were incubated with anti-Alexa Fluor 488 polyclonal antibody (A11094, Thermo) in PBS (1/250) to quench the extracellular Alexa 488 labeling then permeabilized with Triton X-100 0.5% for 3 min prior to DAPI staining. Coverslips were observed under a Zeiss LSM 880 confocal microscope.

**Dynabeads stimulation with a permanent magnet**. Dynabeads were coated with the indicated proteins according to manufacturer's protocol. Cells were grown on 10 cm tissue culture treated dishes for 24 h. Beads were individualized by passing through a 27 G needle and were incubated with the cells for 20 min. Magnetic field was generated by applying a permanent Neodymium magnet on top of the cell monolayer for the indicated time as described in ref. [19] in order to generate a force of 10 pN per bead.

**Traction force microscopy**. Cells were grown on FN-coated polyacrylamide hydrogels of given EM embedded with 1 μm fluorescent microbeads (Matrigen, CA, USA) for 12 h. Briefly, live cells were imaged using an inverted zeiss microscope with controlled temperature and $CO_2$ before and after addition of 0.1% SDS to lyze the cells and relax the gel. TFM was performed in Fiji using the particle image velocimetry (PIV) and Fourier transform traction cytometry plugins described in ref. Tseng et al. (2012)[46] and available at https://sites.google.com/site/qingzongtseng/imagejplugins. At least 5200 force vectors from 3 fields for each condition were calculated. Force field graphs were generated in SigmaPlot. Alternatively, TFM, map generation was also performed using the TFM package developed and distributed by Gaudenz Danuser's lab (UT Southwestern, Dallas, TX, USA)[47].

**Magnetic force assay**. The UNC three-dimensional force microscope was used for applying controlled and precise local force on the magnetic beads (CISMM, UNC Chapel Hill, Chapel Hill, NC, USA). Force calibration was carried out as previously described[19]. Briefly, beads were diluted in a fluid on known viscosity and force was applied to the beads using the magnetic tweezers. Applied force was determined using the equation $F = 6\pi\mu Rv$. Beads displacements were recorded and tracked using Video Spot Tracker in order to determine beads' speed. Cells were plated on FN-coated glass coverslips for 4 h and incubated for 20 min after addition of beads. On force application, bead displacements were recorded with a high-speed video camera (Pulnix, JAI) and tracked using Video Spot Tracker (Center for Computer Integrated Systems for Microscopy and manipulation, http://cismm.cs.unc.edu). Beads coordinates were extracted in Matlab and displacements calculated in Excel. Statistical analysis and plotting were performed in Graphpad Prism6.

**Metabolomics screen**. The metabolomics screen was performed by Metabolon (Durham, NC, USA). Samples were prepared using the automated MicroLab STAR® system from Hamilton Company. Several recovery standards were added prior to the first step in the extraction process for QC purposes. To remove protein, dissociate small molecules bound to protein or trapped in the precipitated protein matrix, and to recover chemically diverse metabolites, proteins were precipitated with methanol under vigorous shaking for 2 min (Glen Mills GenoGrinder 2000) followed by centrifugation. The resulting extract was divided into five fractions: two for analysis by two separate reverse phase (RP)/UPLC–MS/MS methods with positive-ion mode electrospray ionization (ESI), one for analysis by RP/UPLC–MS/MS with negative-ion mode ESI, one for analysis by HILIC/UPLC-MS/MS with negative ion mode ESI, and one sample was reserved for backup. Samples were placed briefly on a TurboVap® (Zymark) to remove the organic solvent. The sample extracts were stored overnight under nitrogen before preparation for analysis. Raw data was extracted, peak-identified and QC processed using Metabolon's hardware and software. Compounds were identified by comparison to library entries of purified standards or recurrent unknown entities. Instrument variability was determined by calculating the median relative standard deviation (RSD) for the internal standards that were added to each sample prior to injection into the mass spectrometers. Overall process variability was determined by calculating the median RSD for all endogenous metabolites (i.e., noninstrument standards) present in 100% of the Client Matrix samples, which are technical replicates of pooled client samples. Values for instrument and process variability meet Metabolon's acceptance criteria and were 4% and 8%, respectively.

The raw intensities of all metabolites in one sample were normalized by dividing through a correction factor. This factor was calculated by dividing the median of all metabolites of one sample through the mean of the medians of all samples. Subsequently, we included a filtering step to remove the least abundant compounds, that were close to the noise level. The normalized intensities were filtered by calculating the mean intensity of each metabolite for all samples. Metabolites with a mean intensity above 1,000,000 were kept. These were in total 447 metabolites. The mean intensity, standard deviation, and coefficient of variation were calculated for each metabolite measured in the four replicates of each condition and are reported in Supplementary Data. The normalized and filtered dataset was the basis to calculate $p$ values ($t$ test) and fold changes between the CD98hc re-expressing CD98hc-null cells and all other conditions. The fold changes were calculated by dividing the means of the corresponding quadruplicate measurements. In order to correct for multiple testing, we used the BH procedure with a false discovery rate set at 5%. The critical values [(rank of the $p$ value/447)×0.05] were calculated and the largest $p$ value which is smaller or equal to the corresponding critical value was determined. For this $p$ value and all smaller $p$ values of the corresponding condition, the null hypothesis was rejected. The rank and critical value are reported in the Supplementary Data 1.

**Statistical analysis**. Cell culture experiments were performed at least three times. For animal experiments, Monte-Carlo simulation was used to determine group size. All animal experiments and RT-qPCR were performed in a blinded fashion. All quantifications represent mean ± standard error of the mean (s.e.m.). Images are representative of experiments that have been repeated at least three times. Group comparison was performed using two-tailed unpaired Student's $t$ test. Multiple groups comparison was performed by two-way ANOVA. Frequency distribution histogram analysis was performed to assess normal distribution. Variance difference was assessed by $F$-test.

## Data availability
The data that support the findings of this study are available from the corresponding author upon reasonable request.

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

## Acknowledgments

The authors thank Lisa Sharek for technical support; Cercina Onesto, Laetitia Seguin, Keith Burridge, Manuel Palacin, Aurélie Rossin, and Anne-Odile Hueber for discussions, advice, and support. The authors thanks Ellen Van Obberghen-Schilling for integrin β3-GFP (INSERM, Nice, France) and Rick Horwitz (UVA, Charlottesville, USA) for integrin α5-GFP. The authors acknowledge Richard Superfine, Tim O'Brien and the Center for Computer Integrated Systems for Microscopy and Manipulation at UNC Chapel Hill (UNC CISMM) supported by the NIH NIBIB (NIH 2-P41-EB002025). The authors acknowledge the IRCAN CytoMed, PICMI, and Animal core facilities which are supported by grants from the Conseil Général 06, the FEDER, GIS IBISA, the Ministère de l'Enseignement Supérieur, the Région Provence Alpes-Côte d'Azur, the Canceropole PACA, the foundation ARC and INSERM. This study was supported by a Marie Curie International Reintegration grant from the European Union Seventh Framework Program under grant agreement #276945 to EB, by grants from the Ligue contre le Cancer (R14035A), la Fondation ARC (R15124AA) and the Société Francaise de Dermatologie (SFD) to C.C.F. F.T. was the recipient of a doctoral fellowship from INSERM Region Provence-Alpes Cote d'Azur Canceropole PACA. The authors' lab was supported by the French Government through the Investments in the Future projects (LABEX SIGNA-LIFE, ref. # ANR-11-LABX-0028-01 and the UCAJEDI ref. # ANR-15-IDEX-01) managed by the National Research Agency (ANR).

## Author contributions

E.B., S.E. and C.C.F. designed, performed experiments, and wrote the manuscript. L.C., F.T., L.R. and L.T. helped with several experimental procedures. S.P. performed the AFM measurements. M.L.H. and A.C.G. analyzed the metabolomics data. C.C.F. directed the project and revised the manuscript. All authors provided detailed comments.

## Additional information

**Competing interests:** The authors declare no competing interests.

