## [Peer Review File · Nature Communications]

Reviewers' comments:

Reviewer #1, an expert in cell adhesion (Remarks to the Author):

Boulter et al are exploring the cross-talk between cell mechanics and cell metabolism by focusing on the regulatory subunit of the heteromeric aminoacid transporter (HAT) system named CD98hc. Based on immortalized dermal fibroblasts from CD98hc dermal conditional Fsp1-Cre, different assays including magnetic force assay using FN-coated beads assay, Rho assay and TFM show that the loss of CD98hc prevents cell stiffening and force generation through RhoA signaling indicating the requirement of CD98hc in the cellular response to mechanical force application on integrin. This correlates with the impairment of stiffness-dependent gene expression controlled by YAP/TAZ (ANKRD1 and CTGF). Whereas integrin engagement is necessary for cell stiffening and RhoA activation, CD98hc is not directly responsible for RhoA activation or cell stiffening as demonstrated by the beads coated by antibodies against CD98hc. CD98hc is also important for proper tissue mechanical homeostasis as shown by AFM and ultrastructural studies. Metabolomic profiling shows that the deletion of CD98hc leads to a defect in sphingolipid metabolism. After quantification and analysis of metabolites, the authors identify that CD98hc deletion leads to a down regulation of the delta-4-saturase DES2 and depletion of DES2 impairs Rho activation.

This study is investigating a cross-talk between cell metabolism and cell adhesion which is an emerging field. Overall, the experiments reported in this study are well performed and controlled and data are novel. However, at this stage the report lacks sufficient mechanistic insights linking integrin and CD98hc/DSE2 and how DSE2 by itself may control mechanosensing. We still do not know whether CD98hc acts as a chaperone or is involved in membrane compartment or membrane composition which might impact on integrin sequestration or integrin internalization. Some issues that should be addressed in order to strengthen the manuscript are listed below.

1. First the introduction needs to be focused on the cross-talk between integrin and metabolism to give a good overview on what it was recently discovered in the field on integrin/metabolism (see for example: Rainero and Norman, Cell report 2015; Cosset and Cheresch, cancer cell 2017; Ata and Antonescu, review 2017...). Furthermore the end of the introduction should recapitulate the main results of the submitted manuscript. Please do not say that environmental parameters are restricted to physical parameters (line 48).
2. Figure 1: The authors have to choose between CD98hc null cells and CD98hc^{-/-} in the figures of the manuscript (choose the same name for all the figures). The authors should assemble Fig 1C and 1D together to better compare rho and rac. The authors need to link CD98hc to b1 integrin associated cell stiffening as claimed in line 149. Indeed b1 and b3 integrin are both important for FN binding. Moreover b3 integrin can compensate. For these reasons it would be important to perform FACS analysis to control cell surface expression of b3 integrin upon CD98hc deletion to complete fig. S1F. In a second step, magnetic beads experiments and RhoA assay need to be performed with and without b1 or b3 integrin in cells deficient or not in CD98hc in order to identify b1 integrin as the unique integrin involved in cell stiffening in both conditions. How to explain the increase of GTP-RhoA in the case of CD98hc deletion at 0 min as compared to 5 min in Fig1E. The authors should also test irrelevant matrix in fig.1E.
3. For the mechanosensing part (Figure 2 and Fig3), the authors need to remove or explain data which are redundant with Estrach et al cancer research paper (2014). There are some discrepancies in fig 2A and 2B. How to explain the increase of cell area between 8kPa and 50kPa in CD98hc cells in the quantification part and the lack of difference in GTP-Rho for the same conditions in fig 2B? TFM images need to be improved (see also fig S4B). Cell borders need to be delimited and magnification should be provided.
4. Fig. S2: The legends of fig S2A do not seem appropriate. Quantification of immunofluorescence would reinforce the message.
5. Fig S4E is not convincing. Legends are missing in Fig. S4I and S4J in order to identify receptors drawn in the schema. Moreover why only sphingolipid metabolism is indicated in the schema?
6. Figure 5: Quantification and analysis of metabolites suggest the involvement of DES2 enzyme. The authors need to validate more the target DES2. Are DSE2 and DES1 sensitive to b1 or b3

integrin deletion? Rho/ROCK activity? siRNA DSE2 and DSE1 need to be performed to check YAP target, LOX expression, GEF-H1 membrane recruitment in order to validate more its action on RhoA activity. Is RhoA still at the membrane if membrane composition is affected? What about FRET analysis and Rho activity?

7. The specificity for b1 integrin is questionable because of GEF-H1 involvement described in Fig 6. The study of b3 integrin is relevant because of its link with GEFH-1 as mentioned by the authors (line 445). GEFH1 is constitutively recruited at the membrane upon CD98 KO. Does it mean that GEF-H1 is not sensitive to mechanical input? How to reconcile with Guilluy data? According Guilluy data, activation of LARG and GEF-H1 involves distinct signaling pathways. LARG is activated by the Src family tyrosine kinase Fyn, whereas GEF-H1 catalytic activity is enhanced by ERK downstream of a signaling cascade that includes FAK and Ras. What is the status of Src and Fyn in term of localization and phosphorylation in case of CD98 deletion? Src kinase activity cannot be studied by following only the status of FAK phosphorylation as shown in Fig. 6B. This approach would complete the section regarding GEF-H1 (Figure 6).

8. The authors need to present a little bit more the characteristics of the CD98hc mutants (C330S, C109S, C98T98E69...). A table might be instructive to better understand the properties of each mutant. Is there any mutant of CD98 unable to interact with CD98 to check the link between CD98, integrin and mechanosensing?

9. The authors need to investigate the status of mTOR. As different examples demonstrate the contribution of mTORC1 in Integrating Amino Acid Sensing (see Ata and Antonescu, review 2017), it would be important to see whether mTOR (or mTOR signaling pathways) is involved or excluded from CD98 axis. Indeed mTORC1 also promotes lipid synthesis by control of Lipin-1 to promote sterol regulatory element-binding protein (SREBP)-dependent transcription (pettersen, cell 2011), and negatively regulates autophagy, such as by phosphorylation of ULK1. Moreover proinvasive integrin trafficking pathway is regulated by nutrient through mTOR signaling (Rainero, Norman, Cell report 2015). If the metabolomic data are not sufficient to rule out mTOR, a simple western blot analyzing mTOR signalling pathway should answer to rule out mTOR signaling.

10. The authors need to deal with the role of CD98/DSE2 in membrane composition or in chaperone function.

11. As mentioned by the authors, the membrane composition might be affected. There is some confusion about the types of membrane. Are the authors talking about ER membrane or about cell membrane? The membrane composition should be analyzed. Would it be possible to localize/stain ganglioside (like in Okhawa, Furukawa et al, JBC 2010) and sphingolipids (see tools in thermofischer, invitrogen). If membrane composition is affected in the case of CD98 deletion, does it change integrin trafficking? Is there any link between Integrin trafficking/mTOR/ autophagy/ lipid (see Tuloup-Minguez cell cycle 2013 and Rainero and Norman, Cell report 2015). At least measurements of integrin sequestration should be provided to see whether the change in membrane composition can affect b1 and b3 integrin dynamics (FRAP, PALM etc..).

12. One of the concerns is the putative role of CD98 as a chaperone of DSE2. No experiments are available in the paper. The authors need to add some data to prove this role. No data justify the role of CD98 as a chaperone.

13. If the authors are talking about DES2 or CD98 as good candidates in the case of aging people (line 392), they need to show staining of DES2 and CD98 in aging skin in order not to extrapolate.

Reviewer #2, an expert in sphingolipid biology (Remarks to the Author):

This manuscript demonstrates that CD98hc is involved in detecting mechanical signals via a cross regulatory mechanism that involves modulating sphingolipid biosynthesis. Based on the experimental results, the authors report that CD98hc senses mechanical signals upstream of RhoA

activation. In their mechanism, CD98hc is involved in the production or stabilization of the sphingolipid biosynthetic enzyme DES2, and they speculate that CD98hc may be involved with mediating the proper folding of DES2. DES2 catalyzes a step in sphingolipid biosynthesis, and sphingolipid levels drop in its absence. Sphingolipid deficiency is accompanied by a cholesterol deficiency, and the two together eliminate cholesterol-enriched microdomains in the plasma membrane, which impedes the GEF-H1 cycling between the plasma membrane and cytosol that is required to activate RhoA.

The proposed cross regulatory mechanism is really interesting, which is a strength of the manuscript. Another major strength of the manuscript is the thorough experimental data and rationale that was presented makes a convincing case that supports most steps in the proposed mechanism. Specifically, the data convinced the reviewer that CD98hc is required for cell response to mechanical signals, CD98hc is upstream of RhoA activation, and RhoA activation isn't defective; CD98hc is involved in cell contractility upstream of RhoA activation and its presence is required for the expression of stiffness-dependent genes; CD98hc was required for normal tissue elasticity, and normal production of 3 collagen-crosslinking enzymes; and CD98hc influences DES2 levels, and this sphingomyelin levels, and both normal sphingolipid levels and CD98hc are required for cell response to mechanical stimuli. Clearly a significant amount of careful planning, expertise, and time was required to complete the wide range of experiments that were used to demonstrate these aspects of the proposed mechanism.

The major weakness is that the experimental evidence more weakly supported the involvement of GEF-H1, and especially cholesterol enriched membrane microdomains in the proposed mechanism. These are the major weaknesses, which are described below.

1. In regard to GEF-H1, the reviewer did not understand how the GST-RhoA pulldown showed GEF-H1 was not functioning properly in the CD98hc null cells, or how the membrane association/cell fractionation experiment showed the GEF-H1 was inactive in the CD98hc cells. Can the authors clarify their explanation of what their findings show and the rationale for how they lead to the conclusion that GEF-H1 was not functioning properly and inactive in the GEF-H1 cells?

2. The involvement of cholesterol enriched membrane microdomains seems to be based on the findings of how sphingolipid levels were correlated with cell response/RhoA activation to mechanical signals, the finding that CD98hc null cells have lower cholesterol levels, and the GEF-H1 from lysed cells is recovered in a membrane fraction when separated on a sucrose gradient. Indeed, the idea that the primary function of cholesterol and sphingolipids in cells is to form special membrane microdomains that recruit a subset of membrane proteins that form complexes and mediate function is a popular hypothesis, and membrane fractionation studies use to be a popular technique for "detecting" them. However, in the past 5 years, a few new complementary high resolution imaging methods have disproved the idea that cholesterol enriched microdomains are present in the membrane- see Egging and coworkers in Nature Comm 2014, 5:5142, Kraft and coworkers in J Biol Chem 2013, 288:16855, and Fujiwara and coworkers in Traffic 2014, 15:583. I do not doubt that CD98hc influences sphingolipid levels, and sphingolipids and possibly cholesterol are involved in CD98hc-mediated mechanical stimuli sensing. However, the data presented in this manuscript does not convince the reviewer that cholesterol enriched membrane microdomains are involved in this mechanism, especially in light of the aforementioned studies that indicate such domains do not exist. Furthermore, the cell lysis and sucrose density separation technique used in this manuscript is no longer considered a reliable method to assess a components intracellular location. At best, it provides information about a component's affinity for membranous material, and the value of that information is questionable. It seems more likely that sphingolipids have some other role in this process. Several studies have shown that sphingolipids can selectively bind to distinct sites on membrane proteins, which modulates their conformation and activity (see Brugger and coworkers in Nature 2012, 481:525 and Barrantes and coworkers in BBA 2009, 1788:2345). Furthermore, sphingomyelin metabolites are signaling molecules, functioning as second messengers that modulate a variety of cell processes; Yusuf Hannun has

published extensively on sphingolipid signaling mechanisms. Either type of mechanism is more plausible than one involving cholesterol enriched microdomains. It seems that the precise mechanism for why sphingomyelin is required for CD98hc-modulated mechanical sensing was not the author's main focus, and cholesterol enriched microdomains were suggested because they are the "go to" explanation for processes that involve cholesterol or sphingolipids. If that is the case, I suggest omitting mentioning cholesterol enriched membrane microdomains and simply say the exact mechanism for why sphingomyelin is required for CD98hc-modulated mechanical sensing will require future studies to elucidate. (I do not think that the mechanism of sphingomyelin involvement needs to be elucidated to warrant publication of this manuscript.) Alternatively, it would be acceptable to mention the other possible mechanisms in addition to cholesterol enriched membrane microdomains to explain the sphingomyelin involvement. If the authors are committed to insist the mechanism involves cholesterol enriched microdomains, the authors would need to simultaneously image GEF-H1 and cholesterol in cells and show they are colocalized in membrane microdomains in normal cells but not CD98hc null cells to provide more convincing support for that aspect of their mechanism.

Reviewer #3, an expert in metabolomics (Remarks to the Author):

Comments on Nat-COM 152284

These comments are on the metabolomics results only.

There are several issues around the metabolomics work. The extraction used is not specific for sphingolipids. There major results need to be confirmed by a method that has been validated for sphingolipid analysis.

It is unclear what the number of samples were that were used in the metabolomics screen. There is no information on the error of the measurement.

There are no details on the data processing of the metabolomics data. From my experience with Metabolon I expect that the method used yielded relative concentrations, that were log transformed and then scaled and the difference expressed as relative fold changes, but this needs to be clarified. With this approach information is lost about the actual difference in concentration between metabolites and over emphasises the difference between low concentration metabolites which measurements are less precise. Again by using a targeted method for sphingolipids it becomes only possible to really assess the effect on the sphingomyelin metabolism.

Minor comments

Metabolon's measurements have a relative standard deviation between 5 and 20%. This means that the numbers given in supplementary table 2 are not presented correctly. With a 10% RSD that means for ketosphinganine the first number of 0.3196 should be shown as 0.3 because the further digits can not be measured. The numbers in the table suggest a level of precision that is not there.

Point by point response to reviewer's comments – NCOMMS-17-33042

Reviewer #1, an expert in cell adhesion (Remarks to the Author):

Boulter et al are exploring the cross-talk between cell mechanics and cell metabolism by focusing on the regulatory subunit of the heteromeric aminoacid transporter (HAT) system named CD98hc. Based on immortalized dermal fibroblasts from CD98hc dermal conditional Fsp1-Cre, different assays including magnetic force assay using FN-coated beads assay, Rho assay and TFM show that the loss of CD98hc prevents cell stiffening and force generation through RhoA signaling indicating the requirement of CD98hc in the cellular response to mechanical force application on integrin. This correlates with the impairment of stiffness-dependent gene expression controlled by YAP/TAZ (ANKRD1 and CTGF). Whereas integrin engagement is necessary for cell stiffening and RhoA activation, CD98hc is not directly responsible for RhoA activation or cell stiffening as demonstrated by the beads coated by antibodies against CD98hc. CD98hc is also important for proper tissue mechanical homeostasis as shown by AFM and ultrastructural studies. Metabolomic profiling shows that the deletion of CD98hc leads to a defect in sphingolipid metabolism. After quantification and analysis of metabolites, the authors identify that CD98hc deletion leads to a down regulation of the delta-4-saturase DES2 and depletion of DES2 impairs Rho activation.

This study is investigating a cross-talk between cell metabolism and cell adhesion which is an emerging field. Overall, the experiments reported in this study are well performed and controlled and data are novel. However, at this stage the report lacks sufficient mechanistic insights linking integrin and CD98hc/DES2 and how DES2 by itself may control mechanosensing. We still do not know whether CD98hc acts as a chaperone or is involved in membrane compartment or membrane composition which might impact on integrin sequestration or integrin internalization. Some issues that should be addressed in order to strengthen the manuscript are listed below.

We acknowledge the reviewer for his comments and insights into our study. The reviewer will find below our point by point response to all the specific points raised. We would like to use this opportunity to discuss and address the main concerns raised by the reviewer in his/her general comments. Beside the specific points discussed below, the reviewer raises three main concerns: the mechanistic link between integrins and CD98hc/DES2, how DES2 controls mechanosensing and whether/how CD98hc acts as a chaperone or affects membrane composition.

First, regarding the link between integrins and CD98hc/DES2, it has previously been established by a wide number of studies (and we did not question nor revisit these published data) that CD98hc associates with both integrins $\beta 1$ and $\beta 3$ (Fenczik et al., 1997), and that, at least for $\beta 1$, CD98hc regulates its function (Fenczik et al., 1997; Feral et al., 2005) (with the exclusion of mechanosensing prior to this manuscript). DES2 has never been associated with integrins to our knowledge, most likely because it is an ER associated protein which catalyzes the conversion of dihydroceramide into ceramide (Fabrias et al., 2012). Here, in the context of this regulation, we have no evidence indicating that CD98hc or DES2 directly regulate integrins in order to control mechanosensing. Indeed, we rather have evidence to conclude that this regulation occurs independently of direct integrin regulation.

- We found that loss of CD98hc does not affect cell surface expression of integrin subunits.
- Mechanical force application on integrins in CD98hc null cells results in the activation of ERK1/2 indicating that integrin mechanosensing per se is not affected, at least down to ERK.

- The CD98hc C330S mutant which also alters mechanosensing, has never been associated with defective integrin signaling and binds integrins properly (Estévez et al., 1998). Also, addition of C12SM is sufficient to restore mechanosensing in C330S expressing cells indicating that sphingolipid depletion is the sole defect preventing proper mechanosensing.

Second, regarding the control of mechanosensing by DES2, we link DES2 to mechanosensing via the synthesis of sphingolipids, independently of integrins. Using metabolomics screening, we initially identified sphingolipid synthesis as a potential regulator of mechanosensing. We show that affecting sphingolipid content, or cholesterol, actually impairs mechanosensing (Fig 4F). We have now characterized the consequences of DES2 depletion by RNA interference, as suggested by the reviewer, and we observed that DES2 depletion also affects integrin-dependent mechanosensing, i.e. RhoA activation, SFK phosphorylation, GEF-H1 localization, Cholesterol content/localization and activation as well as YAP/Taz activation. We hope this complete series of experiment help clarify how DES2 controls mechanosensing, i.e. by controlling sphingolipid availability. As discussed with reviewer #2, sphingolipids may then regulate mechanosensing either via second messengers or membrane microdomains.

Finally, regarding the regulation of DES2 levels by CD98hc, we initially hypothesized that CD98hc may behave as a molecular chaperone for DES2 largely based on previous papers reporting CD98hc to behave as such for other transmembrane proteins such as LAT1 (Cormerais et al., 2016). This was also supported by the fact that DES2 level in cells was sensitive to molecular chaperone inhibition. We have now investigated this possibility further and come up with a more detailed picture. Please keep in mind that DES2 is a multiple span transmembrane protein which localization is restricted to the ER where it converts dihydroceramide into ceramide (Fabrias et al., 2012). Now, we show, by co-immunoprecipitation, that in CD98hc null or C330S cells, in which DES2 is downregulated, DES2 constitutively associates with molecular chaperones such as Grp94, hsp90 counterpart in the ER, while it does not in control cells. This seemingly indicates that DES2 folding becomes problematic in the absence of CD98hc or upon expression of C330S mutant and requires intervention of molecular chaperones. Indeed, treatment of control cells with geldanamycin an hsp inhibitor, which downregulates DES2 levels, results in the increased association of grp94 with DES2 to levels similar to that of untreated CD98hc null cells. Additionally, we also observed that grp78, hsp70 counterpart, is recruited to the complex in CD98hc null and C330S cells. Despite numerous attempts, we were never able to detect any association of DES2 with CD98hc under any circumstances which may either indicate that they do not bind each other or that this association is very labile. Both DES2 and CD98hc are transmembrane proteins shuttling through the ER at some point during their synthesis. Throughout this process, CD98hc C330 residue would lie in the ER lumen, on the same side of the membrane as the two small ER lumen facing loops of DES2, none of which contains a cysteine. Altogether, this seemingly indicates that a disulfide bridge is unlikely to exist between CD98hc C330 and DES2. These data point out that CD98hc is somehow required for proper folding of DES2 in the ER and that loss of CD98hc triggers activation of the molecular chaperone system in order to attempt to salvage DES2. At this time, we do not know how, at the molecular level, CD98hc affects DES2 folding. This is an interesting molecular mechanism we seek to investigate further. However, we feel this lies beyond the scope of this manuscript focused on the cross-talk regulation of mechanosensing and sphingolipid metabolism.

1. First the introduction needs to be focused on the cross-talk between integrin and metabolism to give a good overview on what it was recently discovered in the field on integrin/metabolism (see for example: Rainero and Norman, Cell report 2015; Cosset and Cheresh, cancer cell 2017; Ata and Antonescu, review 2017...).

Furthermore the end of the introduction should recapitulate the main results of the submitted manuscript. Please do not say that environmental parameters are restricted to physical parameters (line 48).

We agree with the reviewer that the introduction should to be improved regarding the recent developments on the cross-talk between integrins and cell metabolism. We would nevertheless like to point out that despite several papers reporting a link between integrins and cell metabolism, none of them investigate mechanosensing. The introduction has now been modified according to these comments to include references to the suggested articles and work mentioned by the reviewer, as well as others. We also modified the introduction in order to recapitulate the main results of the manuscript. Additionally, the statement regarding environmental parameters and physical parameters has been reformulated.

2. Figure 1: The authors have to choose between CD98hc null cells and CD98hc^{-/-} in the figures of the manuscript (choose the same name for all the figures). The authors should assemble Fig 1C and 1D together to better compare rho and rac. The authors need to link CD98hc to b1 integrin associated cell stiffening as claimed in line 149. Indeed b1 and b3 integrin are both important for FN binding. Moreover b3 integrin can compensate. For these reasons it would be important to perform FACS analysis to control cell surface expression of b3 integrin upon CD88hc deletion to complete fig. S1F. In a second step, magnetic beads experiments and RhoA assay need to be performed with and without b1 or b3 integrin in cells deficient or not in CD98hc in order to identify b1 integrin as the unique integrin involved in cell stiffening in both conditions. How to explain the increase of GFTP-RhoA in the case of CD98 deletion at 0 min as compared to 5 min in Fig1E. The authors should also test irrelevant matrix in fig.1E.

We apologize for this inconsistency in figure labelling. We acknowledge the reviewer for bringing it up to our attention and giving us an opportunity to fix this. The figures have now been proofed and modified for labeling consistency wherever required. Briefly, we used the labelling "CD98hc null" for immortalized fibroblasts genotyped as Fsp1-CRE CD98hc fl/fl and we precisely indicate the genotype when referring to animals.

The figure fusion suggested by the reviewer is not clear for us. Fig1C and 1D do not address the activation of Rac but RhoA, respectively in a time course assay and a CD98hc rescue experiment. Is the reviewer referring to Fig1G? Does he suggest a fusion of former Fig1C and 1G (now 1C and 1H)?

The reviewer raises a number of interesting points on integrin $\beta 3$. In light of the function of integrin $\beta 3$ in mechanosensing (as described in (Schiller et al., 2013)), we agree that the role of integrin $\beta 3$ in that regulatory process could be explored and we would like to use that opportunity to clarify our perspective. Originally, in the manuscript, we reported the result of the mechanical stimulation of control or CD98hc null fibroblasts using either FN- or Coll-coated magnetic beads on RhoA activation, which yields to activation of RhoA in control cells and defective activation in CD98hc null cells. We therefore concluded that "Altogether, this seemingly indicates that CD98hc regulates mechanosignaling to RhoA downstream of integrin $\beta 1$, the receptor for these two ECM proteins". We did not intend to mean that integrin $\beta 3$ was irrelevant to this process nor question published results such as those of Schiller et al. Instead, independently of integrin $\beta 3$ and according to Guilluy and Burrridge, we intended to confirm that this regulation was associated with integrin $\beta 1$ on the basis of the ECM proteins used to

coat the beads which both require $\beta 1$ containing receptors. Additionally, type I collagen is abundant in the dermis and it is therefore physiologically relevant to assess if that regulation occurs with collagen in these dermal cells. Altogether, this justified our claim for the specificity for integrin $\beta 1$ and our decision to focus on it as also questioned on specific point #7. Also, we would like to emphasize that the focus of this manuscript is not to investigate the respective contribution of each integrin to mechanical signaling, which has been done before much more thoroughly by others in dedicated studies, but to assess the crosstalk between cell metabolism and integrin mechanosensing.

Nevertheless, following the reviewer's comment and in order to take integrin $\beta 3$ into account, we measured the cell surface expression of both integrins $\beta 3$ and αv (each of which constitute a subunit of the $\alpha v\beta 3$ dimer, another receptor for FN) in the immortalized control and CD98hc null dermal fibroblasts used in the present study. We did not observe any modification of the cell-surface expression of integrin $\beta 3$ in these cells. This data was added to Supplemental Figure 1F. We also observed no alteration of the expression of integrin αv at the cell surface. This piece of data was not included in the manuscript because the original electronic data was compromised upon a hard drive crash and we only have printed integrin αv expression profiles. These are included as a reviewers-only figure appended to this point by point response (Reviewers Figure 4). Additionally, we also documented the activation status of integrin $\beta 1$ in these cells using the conformation specific 9EG7 antibody, which has also been included in Supp. Fig1. We observed no difference in integrin $\beta 1$ activation. We extended that study to assess whether affecting $\beta 3$ function would alter mechanosensing in the conditions used in that study. We blocked integrin $\beta 3$ using function-blocking anti- $\beta 3$ antibody LM609 on HeLa cells, for which we have performed similar experiments as those on fibroblasts (included in the manuscript) and because this antibody is specific to human cells. We incubated HeLa cells with LM609 at a concentration of $25\mu\text{g}/\text{mL}$ for 30 minutes prior to addition of FN-coated magnetic beads and mechanical stimulation. We observed that, in that particular experimental design, blocking integrin $\beta 3$ had no effect on mechanical activation of RhoA. This data has been added as Supp. Fig1I. As a note, depletion of DES2 does not affect integrin expression either (Fig S6).

In his comments, the reviewer suggests an elegant experiment, to suppress expression of integrin $\beta 1$ or $\beta 3$ and assess their respective contribution to mechanosensing. Although appealing conceptually, this experiment is practically difficult to perform and interpret. Indeed, silencing the expression of integrin $\beta 3$ and, even more, that of integrin $\beta 1$ would result in the invalidation of the associated receptors, namely, in these cells, the vitronectin/fibronectin receptor $\alpha v\beta 3$ and /or the $\beta 1$ -based integrin receptors such as the FN receptor $\alpha 5\beta 1$ and the Coll receptor $\alpha 1\beta 1$. This would result in one of two things: first, the depleted receptor is expressed in very large amounts at cell surface and is mostly responsible for the binding of the magnetic beads, its reduction will largely impair the ability of cell to bind the magnetic beads. Upon application of the magnetic field, beads will literally fly out to the magnet if unbound or will be ripped off the cells upon weak binding. For instance, we previously coated magnetic beads with anti-CD98hc RL388 antibody to assess whether CD98hc is a mechanosensor. Beside using these beads with control fibroblasts (Fig1I), we used these beads with CD98hc null cells and observed that the beads could not bind to the cells. They were simply sitting atop the cells and went flying when applying the magnetic field. Therefore, they did not apply any stimulation to the cells. We anticipate the same may happen upon depletion of $\beta 1$ or $\beta 3$. Second option, the targeted receptor is not essential for the binding of the beads and the beads may still bind to the cells despite its loss. Nevertheless, the composition of adhesion complexes will be modified, and the amount of targeted receptor will be reduced resulting in less receptors bound to a single bead. One has to keep in mind that the force generated by a single bead is a function of its radius and the intensity of the magnetic field, since these two factors are constant, the same force is

generated by every bead. However, at the level of the adhesion complex, the force applied by the bead is divided equally between each bound receptor. Therefore, if the number of receptors is affected, the amount of force applied to each remaining receptor will be drastically different. In that case, the stimulation will be different between control and integrin-depleted conditions preventing us from drawing any conclusion.

Therefore, we did not perform the suggested experiment to silence $\beta 1$ or $\beta 3$ by siRNA and assess RhoA activation upon mechanical force application using magnetic beads due to technical and experimental considerations. Nevertheless, integrin $\beta 1$ requirement in mechanosensing has previously been addressed by Guilluy and Burridge using function-blocking antibody (Guilluy et al., 2011) and we addressed integrin $\beta 3$ requirement by using function-blocking LM609 antibody as mentioned earlier.

Regarding the activation of RhoA with coll-coated beads, we have observed this increased level of GTP-RhoA upon addition of Coll-coated beads without application of magnetic field in several instances. At this point, we have no molecular explanation nor experimental data to account for that effect. Nevertheless, one should keep in mind that the design of the experiment highlights two aspects of integrins function: mechanosensing upon application of the magnetic field, which the experiment has been designed for, and incidentally, integrin ligation without mechanical stimulation at $t=0$. This is precisely what is observed here. Since CD98hc regulates several aspects of integrin function, it is not necessarily unexpected to see differences in GTP-RhoA levels upon integrin ligation. For instance, we previously reported that loss of CD98hc affected RhoA activation under several circumstances including adhesion on FN or wound healing in vivo which may relate more closely to Coll (Boulter et al., 2013; Feral et al., 2007). At this stage, we have no explanation for this difference in RhoA activation, however, since the scope of our manuscript is the crosstalk regulation between cell metabolism and integrin mechanosensing, we did not investigate further this difference in RhoA activation upon ligation of Coll-coated beads.

The experiment suggested by the reviewer to assess the effect of irrelevant matrix although conceptually appealing, is practically difficult to perform for reasons similar to that of the integrin silencing experiment: if we coat the beads with some irrelevant matrix (ie matrix protein with no receptor on the cells), the beads will not bind to the cells and no force can therefore be applied. For instance, we previously tested BSA-coated beads that would not bind to the cells and therefore, did not stimulate mechanosensing.

3. For the mechanosensing part (Figure 2 and Fig3), the authors need to remove or explain data which are redundant with Estrach et al cancer research paper (2014). There are some discrepancies in fig 2A and 2B. How to explain the increase of cell area between 8kPa and 50kPa in CD98 cells in the quantification part and the lack of difference in GTP-Rho for the same conditions in fig 2B? TFM images need to be improved (see also fig S4B). Cell borders need to be delimited and magnification should be provided.

We acknowledge the reviewer for pointing that out, but no data is actually replicated from the Estrach 2014 Cancer Research paper. This Cancer Research paper investigated the role of CD98hc in tumorigenesis in the context of epithelial cancer. Therefore, CD98hc was genetically invalidated specifically in keratinocytes in the epidermis using a K14 CRE:ER mouse model. All experiments were therefore carried out either using CD98hc deficient keratinocytes and mouse models deficient in epidermal CD98hc. This is radically different from the current manuscript which investigates the role of CD98hc in mechanosensing in dermal fibroblasts. As the

reviewer is most likely aware, although part of the same organ, epidermis and dermis are completely different tissues with different roles and properties (epithelial vs mesenchymal, ECM poor vs ECM enriched,...). Therefore, all experiments reported in the Cancer Research paper were carried out on keratinocytes or the epidermis while all experiments reported in the current manuscript were performed on dermal fibroblasts and the dermis. In addition, in the Cancer research paper, we used CD98hc null MEFs in order to document FAK and AKT phosphorylation upon adhesion of FN-coated PDMS hydrogels. Neither do the experimental set up nor the cell model overlap with our current manuscript, but those results nevertheless support our current findings since src-mediated FAK phosphorylation as well as cell spreading were impaired in CD98hc null MEFs. Additionally, we explore in the current manuscript a novel crosstalk regulation that was neither anticipated nor investigated at all in the Cancer Research paper.

We appreciate the reviewers' thoroughness and his interesting point regarding the relationship between RhoA and cell spreading. The wide-spread belief is that RhoA tightly regulates cell spreading such as that any alteration of RhoA activity would alter cell spreading. This dogma stems from the early work on cell adhesion by several labs including Keith Burridge's showing that during cell spreading on fibronectin, RhoA was transiently inhibited before full activation, generation of stress fibers and cell spreading (Arthur and Burridge, 2001; Arthur et al., 2000). However, this dogma is probably not so straightforward. The first author of the present manuscript was a postdoc with Keith Burridge and spent a large amount of time at investigating and manipulating Rho proteins level (Boulter et al., 2010). We developed tools to manipulate RhoA levels such as a RhoA miR shRNA which is co-cistronically expressed with a GFP marker protein (Aghajanian et al., 2009). Expression of this RhoA shRNA in HeLa cells decreases RhoA expression to virtually undetectable levels as illustrated in the appended figure for the reviewers (Fig1). Nevertheless, this very low level of RhoA in no way alters the ability of cells to spread on fibronectin as compared to control cells. This indicates that despite the wide-spread belief that RhoA is essential for cell spreading, alteration of RhoA expression/activity does not affect cell spreading even at early stages of cell spreading. We think a similar regulation occurs on hydrogels and explains why variations in RhoA activation do not translate into modifications of cell spreading. Therefore, RhoA activation and cell spreading are uncoupled which explains why RhoA activity and cell spreading are not coordinated in that experiment.

We apologize for the small size of the TFM images which were due to the figure layout and space constraints. We have replaced these images with new images which both depict traction force field, traction vectors and cell overlay. These were generated using Matlab and a TFM package from Gaudenz Danuser's lab as indicated in the method section and as referred to.

4. Fig. S2: The legends of fig S2A dos not seem appropriate. Quantification of immunofluorescence would reinforce the message.

As advised by the reviewer, quantification of the nuclear localization of YAP has been added to Fig S2. Briefly, we categorized cells in 2 groups based on YAP localization: cytosolic and nuclear, and exclusively nuclear. We counted at least n=100 cells in each condition from n=2 experiments, grouped and plotted the percentage of cells in each group for each condition.

5. Fig S4E is not convincing. Legends are missing in Fig. S4I and S4J in order to identify receptors drawn in the schema. Moreover why only sphingolipid metabolism is indicated in the schema?

We are not sure to understand why and how the reviewer does not find Fig S4E convincing. We assume, one issue may stem from the labelling of the panel which, we realized upon the reviewer's comment, may be ambiguous. Briefly, the purpose of Fig S4E was to show that addition of C12SM does not rescue RhoA activation by mechanical signals in CD98hc null cells. Indeed, we did not observe any activation of RhoA in the condition labelled "control" which is indeed CD98hc null and not "control" as used elsewhere in the manuscript to label CD98hc expressing cells. We used the label control here as opposed to C12SM treatment but in both cases the cells were CD98hc null. We realized this labelling may confuse the reader and we have now modified it to prevent this ambiguity by replacing "control" by "Vehicle". Upon addition of C12SM on CD98hc null cells, RhoA activation downstream of mechanical force application on integrins is not restored.

We apologize for the lack of information regarding the receptors drawn in the figure, this was an omission. We have modified the figure and legend in order to identify each receptor.

In the schema, only sphingolipid metabolism was indicated for the sake of clarity and due to lack of space. We provide, appended to this point by point response, the same schema identifying all the metabolic pathways displayed (reviewer Figure 3). If the reviewer believes that information should be added to figure S4 for the reader's sake, we propose to include it as a supplementary figure with numbered pathways referred to in the legend.

6. Figure 5: Quantification and analysis of metabolites suggest the involvement of DES2 enzyme. The authors need to validate more the target DES2. Are DSE2 and DES1 sensitive to b1 or b3 integrin deletion? Rho/ROCK activity? SiRNA DSE2 and DSE1 need to be performed to check YAP target, LOX expression, GEF-H1 membrane recruitment in order to validate more its action on RhoA activity. Is RhoA still at the membrane if membrane composition is affected? What about FRET analysis and Rho activity?

We acknowledge the reviewer for this suggestion which provided interesting results as explained just below. As suggested by the reviewer, we have performed a number of experiments to validate DES2 as an important regulator of the process described in this manuscript. This lead us to generate a new figure that intercalates between former figures 5 and 6. Briefly, we have generated dermal fibroblasts depleted in DES2 by expression of DES2-specific shRNAs and assessed a number of features we previously reported for CD98hc. From these experiments, it turns out that while it is very clear that DES2 regulates some of them in a similar manner as CD98hc does, others are more remotely connected to DES2. First, we have verified that shRNA-mediated DES2 depletion was effective (Fig 6B) and that it did not affect cell surface expression of the integrin repertoire (Supp Fig 6). Depletion of DES2 decreased RhoA activation, SFK phosphorylation at Y416, GEF-H1 activation in response to mechanical forces application, and increased GEF-H1 localization at cell membrane much like what we observe in CD98hc null cells. We also monitored YAP/Taz activation by RT-qPCR when growing cells on hydrogels and found out that DES2 silencing reduced the YAP/Taz activation response to mechanical cues, however, not to the same extent as CD98hc depletion. Nevertheless, this reduced inhibition of YAP/Taz has to be

put into perspective with the depletion of DES2 which is only partial with the use of siRNA in contrast to the genetic invalidation of CD98hc which is total. Altogether, these results indicate that, as proposed earlier, DES2 is required for the activation of RhoA downstream of force application on integrins as its depletion blocks activation of its upstream regulators SFK and GEF-H1 similarly as CD98hc depletion does. Along this line, it seems that a direct read out of RhoA activation, YAP/Taz dependent transcription, is also affected. However, we also performed TFM on these cells which gave quite unexpected results. We measured traction forces generated by both cells on 8kPa FN-coated hydrogels

The question as to whereas RhoA membrane localization is affected by membrane composition is difficult to address and answer. RhoA membrane localization is tightly coupled to its activation: when GDP-RhoA is sequestered by RhoGDI in the cytosol while GTP-RhoA has been released from RhoGDI prior or during its activation and localizes to the membrane thanks to its prenylation and its adjacent polybasic region. Also, because of this prenyl moiety, RhoA cannot fold properly in solution in the cytosol and is targeted for degradation whenever it is not bound to RhoGDI or a membrane. Hence, it is difficult to assess RhoA localization in control or CD98hc null cells with our assay because RhoA activity itself is affected. One might expect a cytosolic localization of RhoA in CD98hc null cells because of its lack of activation upon mechanical stimulation. Indeed, we observe that much more RhoA is recruited at the membrane in control cells than in CD98hc null cells, it is however not possible to conclude whether this is due to the difference in RhoA activation or to an effect of membrane composition.

7. The specificity for b1 integrin is questionable because of GEF-H1 involvement described in Fig 6. The study of b3 integrin is relevant because of its link with GEFH-1 as mentioned by the authors (line 445). GEFH1 is constitutively recruited at the membrane upon CD98 KO. Does it mean that GEF-H1 is not sensitive to mechanical input? How to reconcile with Guilluy data? According Guilluy data, activation of LARG and GEF-H1 involves distinct signaling pathways. LARG is activated by the Src family tyrosine kinase Fyn, whereas GEF-H1 catalytic activity is enhanced by ERK downstream of a signaling cascade that includes FAK and Ras. What is the status of Src and Fyn in term of localization and phosphorylation in case of CD98 deletion? Src kinase activity cannot be studied by following only the status of FAK phosphorylation as shown in Fig. 6B. This approach would complete the section regarding GEF-H1 (Figure 6).

We acknowledge the reviewer for his insights on GEF-H1 and integrins. The specificity for integrin b1 is discussed in the specific point #2 on integrins. Our understanding of the regulation of GEF-H1 by mechanical signal is mainly based on the studies of Schiller and Fassler (Schiller et al., 2013) and from Guilluy and Burridge (Guilluy et al., 2011). From their work, we understand that GEF-H1 is recruited to the adhesion complex by b3 integrins while integrin b1 functions as a mechanosensor to trigger activation of GEF-H1 via ERK1/2. We agree that we did not assess integrin b3 function in the crosstalk regulation described here and that aspect is being discussed in the specific point about integrins. Nevertheless, we do not think our conclusions contradict the findings of these two aforementioned studies. We believe that GEF-H1 constitutive recruitment to the membrane in CD98hc null cells (Fig 6C) results from a failure to be released because of the defect in sphingolipid synthesis and lipid raft formation. This prevents GEF-H1 from being activated by mechanical forces applied on integrins (Fig 6A). This does not imply that GEF-H1 is not activated by mechanical signals in control cells as originally demonstrated by Christophe Guilluy (Guilluy et al, NCB, 2011). Indeed, it is activated by mechanical signals as we observe in control cells (Fig.6A).

Overall, based on our data, we envision that GEF-H1 does not get activated by mechanical forces in CD98hc null cells because it is sequestered at the plasma membrane, consequently to the loss of lipid rafts (Fig.6G and H; Fig7 G) triggered by loss of sphingolipids and DES2 expression.

To carry on with the work of Guilluy and Burrige, we observe that both LARG and GEF-H1 are activated by mechanical forces in control cells (Fig.6A) while both GEFs are no longer activated in CD98hc null cells. We confirm the existence of these independent pathways for each GEF. Regarding the GEF-H1 pathway, we observe that while GEF-H1 is not activated by mechanical forces in CD98hc null cells, its upstream activator, ERK does get activated (Fig.6B) indicating that the signal is somehow disturbed between ERK and GEF-H1. Regarding the LARG pathway, we observe that both LARG and its upstream activators Src kinases are no longer activated by mechanical forces in CD98hc null cells indicating that signaling is perturbed upstream or at the level of Src kinases and LARG itself.

We agree with the reviewer that Src kinase activation would benefit from additional documentation. We previously reported that SFK activation was generally impaired in CD98hc null cells or tissues (J Exp Med). We have now investigated the phosphorylation of SFK on tyrosines 416 and 527 which are respectively activating or inhibiting its activity, upon application of mechanical forces on integrins.

8. The authors need to present a little bit more the characteristics of the CD98hc mutants (C330S, C109S, C98T98E69...). A table might be instructive to better understand the properties of each mutant. Is there any mutant of CD98 unable to interact with CD98 to check the link between CD98, integrin and mechanosensing?

We understand that the different mutants and their characteristics may be confusing, and we agree with the reviewer that a table would help recapitulate their properties. We generated a table that lists the various mutants and chimeras used in the study, and their characteristics based on the references listed with the table.

We are not sure to understand the last sentence in the reviewer's comment. Does the reviewer actually refer to homo-oligomerization of CD98hc? In such case, not much is known about it. CD98hc homo-oligomerization has long been hypothesized but never satisfactorily addressed. Several articles report that CD98hc clustering using antibodies might participate to its activation but it is not very clear what such an "activation" actually means and no data at the molecular level supports this hypothesis. No CD98hc mutant has ever been reported to block homo-oligomerization because such oligomerization has never been characterized at the molecular level.

9. The authors need to investigate the status of mTor. As different examples demonstrate the contribution of mTORC1 in Integrating Amino Acid Sensing (see Ata and Antonescu, review 2017), it would be important to see whether mTOR (or mTOR signaling pathways) is involved or excluded from CD98 axis. Indeed mTORC1 also promotes lipid synthesis by control of Lipin-1 to promote sterol regulatory element-binding protein (SREBP)-dependent transcription (pettersen, cell 2011), and negatively regulates autophagy, such as by phosphorylation of ULK1. Moreover proinvasive integrin trafficking pathway is regulated by nutrient through mTor signaling (Rainero, Norman, Cell report 2015). If the metabolomic data are not sufficient to rule out mTOR, a simple western blot analyzing mTOR signalling pathway should answer to rule out mTOR signaling.

We acknowledge the reviewer for his comments and his broad vision of the interactions between cell adhesion and other cellular processes. As mentioned by the reviewer, mTOR is widely recognized as an essential regulator and sensor of amino acid availability. As we indicate in the introduction, CD98hc is the heavy regulatory subunit of

the HATs and controls transport of several amino acids across the plasma membrane which constitutes a good candidate to functionally interact with mTOR. Indeed, CD98hc and mTOR have functionally been associated with each other (Nicklin et al., 2009), namely CD98hc and mTORC1, which constitutes a rationale to investigate the relationships between CD98hc and mTOR during mechanosensing.

Originally, several pieces of evidence presented in our manuscript prompted us to dismiss a possible regulation of integrin mechanosensing by aminoacids and mTOR in this context: among these, the C330S mutant used in this study disrupts mechanical signaling from integrins to RhoA but has been characterized as unable to affect aminoacid transport (Estévez et al., 1998). Along this line, metabolomics data gathered and presented in this manuscript indicated that aminoacid metabolism was not affected by expression of this mutant.

As mentioned by the reviewer, mTORC1 promotes lipid synthesis and has been associated with CD98hc making it a good candidate to assess. In order to experimentally address the role of mTOR in the CD98hc-dependent regulation of integrin mechanosensing, we stimulated control cells in the presence of the classic mTORC1 inhibitor rapamycin. We treated cells with 100nM rapamycin for one hour prior to addition of FN-coated beads and mechanical stimulation. We observed that inhibition of mTOR does not affect the mechanical activation of RhoA in dermal fibroblasts. This has been added as panel G in Figure 1.

10. The authors need to deal with the role of CD98/DSE2 in membrane composition or in chaperone function.

This point and specific point #12 both rightfully question how CD98hc regulates DES2 level. We initially stated that CD98hc may behave as a chaperone for DES2. This was based on published data showing that CD98hc does regulate the expression/stability of some transmembrane proteins such as LAT1 and on experimental data presented in the manuscript showing that loss of CD98hc resulted in reduced DES2 levels and that DES2 required molecular chaperones for proper folding. We have carried on these investigations and we have found, by coimmunoprecipitation experiments, that DES2 associates constitutively with grp94, hsp90 counterpart in the ER, upon loss of CD98hc or expression of the C330S mutant (Fig 5). This does not occur in control cells. This indicates that in the absence of CD98hc or expression of the C330S mutant, DES2 folding in the cell becomes problematic and triggers the intervention of molecular chaperones of the hsp family. More, inhibiting proper DES2 folding by blocking hsp90-family proteins with geldanamycin resulted in the cumulative association of DES2 with grp94 in control cells. Additionally, we found that only in CD98hc null or C330S cells DES2 also associates with ER Hsp70 counterpart Grp78.

11. As mentioned by the authors, the membrane composition might be affected. There is some confusion about the types of membrane. Are the authors talking about ER membrane or about cell membrane? The membrane composition should be analyzed. Would it be possible to localize/stain ganglioside (like in Okhawa, Furukawa et al, JBC 2010) and sphingolipids (see tools in thermofischer, invitrogen). If membrane composition is affected in the case of CD98 deletion, does it change integrin trafficking? Is there any link between Integrin trafficking/mTOR/autophagy/ lipid (see Tuloup-Minguez cell cycle 2013 and Rainero and Norman, Cell report 2015). At least measurements of integrin sequestration should be provided to see whether the change in membrane composition can affect b1 and b3 integrin dynamics (FRAP, PALM etc..).

We apologize to the reviewer if we were not clear about which membrane we were referring to. Generally speaking, we refer to the plasma membrane when we mention membrane. Nevertheless, some distinction and

precision can be made. As mentioned earlier, DES2 is strictly restricted to the ER where it converts dihydroceramide into ceramide which is then transported to the golgi by specialized proteins such as CERT. Therefore, when we mention DES2, we refer to the ER. However, when we mention sphingolipids, we refer to the plasma membrane where sphingomyelins and derivatives reside.

Regarding our experimental data, we refer to membrane, as a generic term, when we describe the membrane fractionation experiments (Fig 6) because the experiment was performed to separate membrane fraction from cytosolic fraction but does not discriminate between different cellular membranes. This would require additional iodixanol gradient separation (Boulter et al., 2010) which was not performed in that study.

Membrane composition has been analyzed indirectly by the metabolomics screening. It gives us the relative content of a number of lipids including membrane constituents such as phosphatidyl lipids and sphingolipids for instance in CD98hc null cells as compared to control cells. This indicates that phosphatidyl lipids are not affected by loss of CD98hc while sphingolipids are. Additionally, we have now quantified sphingolipid content which indicates a 25% reduction in sphingolipid content in CD98hc null cells as compared to control cells. Also, cholesterol content has been quantified.

We agree with the reviewer that it is possible that loss of CD98hc or modification of membrane composition may affect integrin trafficking. However, here we provide no evidence that integrin function is affected by loss of CD98hc. It has previously been shown that integrin activation, as measured by 9EG7, is not affected by loss of CD98hc and we show that integrin mechanosensing, strictly speaking, is not affected by loss of CD98hc since ERK1/2 get activated. Therefore, there is no experimental data that would prompt us to suspect an alteration of integrin function. So, while such a study would be interesting from a general standpoint, we have no experimental or conceptual reason to suspect that integrin trafficking would be involved in the mechanism described in this manuscript. Additionally, such investigation would constitute a whole independent study by itself which is far beyond the scope of our manuscript.

Finally, we did not stain cells for gangliosides but instead as suggested by reviewer #2, an expert in sphingolipids, we have simultaneously stained cells for cholesterol and GEF-H1. The results of this staining are now presented in figures 7, S7 and S8. The results are discussed in the text and with more details in the point by point response to reviewer #2. Very briefly, we observed some colocalization of cholesterol and GEF-H1 only in control cells and overall, in CD98hc null cells, we observed that cholesterol staining was much lower and that GEF-H1 was sequestered in the plasma membrane according to the model proposed in the manuscript.

12. One of the concerns is the putative role of CD98 as a chaperone of DSE2. No experiments are available in the paper. The authors need to add some data to prove this role. No data justify the role of CD98 as a chaperone.

We previously answered to a similar comment in specific point #10. We show in that manuscript that DES2 is instable and requires Hsp to be folded properly. We also show that loss of CD98hc results in decreased expression of DES2 that cannot be accounted for by transcriptional regulation since DES2 mRNA is not affected. Since CD98hc has been shown to function as a chaperone for other transmembrane proteins such as LAT1, we hypothesized that the same may occur with DES2. Indeed, after additional investigations we have a different vision now. We show by coimmunoprecipitation experiments, that DES2 associates constitutively with grp94, hsp90 counterpart in the ER, upon loss of CD98hc or expression of the C330S mutant (Fig 5). This does not occur in control cells. This indicates that in the absence of CD98hc or expression of the C330S mutant, DES2 folding in the cell becomes problematic and triggers the intervention of molecular chaperones of the hsp family.

More, inhibiting proper DES2 folding by blocking hsp90-family proteins with geldanamycin resulted in the cumulative association of DES2 with grp94 in control cells. Therefore, we envision that CD98hc is not a molecular chaperone like Hsp but is nevertheless required for proper folding and expression of DES2.

13. If the authors are talking about DES2 or CD98 as good candidates in the case of aging people (line 392), they need to show staining of DES2 and CD98 in aging skin in order not to extrapolate.

We disagree with the reviewer on that statement because we never claimed or mentioned the possibility that DES2 may be a good candidate during aging. Precisely, on line 392, we state that “We show this regulation has biological consequences at the cellular level as well as at the tissue level since invalidation of CD98hc in the dermis results in aberrant skin mechanical homeostasis much like what happens during aging.” We only claim that loss of CD98hc in the dermis results in modifications/degradation of the mechanical properties of the skin that parallel those occurring during aging (Achterberg et al., 2014). Indeed, we reported, independently of this study, that CD98hc expression decreases during aging (Boulter et al., 2013). We added a sentence to mention this “This is not completely surprising since CD98hc expression decreases during aging in skin”. We did not at any point link DES2 to aging and we don’t know how DES2 expression changes during aging.

Reviewer #2, an expert in sphingolipid biology (Remarks to the Author):

This manuscript demonstrates that CD98hc is involved in detecting mechanical signals via a cross regulatory mechanism that involves modulating sphingolipid biosynthesis. Based on the experimental results, the authors report that CD98hc senses mechanical signals upstream of RhoA activation. In their mechanism, CD98hc is involved in the production or stabilization of the sphingolipid biosynthetic enzyme DES2, and they speculate that CD98hc may be involved with mediating the proper folding of DES2. DES2 catalyzes a step in sphingolipid biosynthesis, and sphingolipid levels drop in its absence. Sphingolipid deficiency is accompanied by a cholesterol deficiency, and the two together eliminate cholesterol-enriched microdomains in the plasma membrane, which impedes the GEF-H1 cycling between the plasma membrane and cytosol that is required to activate RhoA.

The proposed cross regulatory mechanism is really interesting, which is a strength of the manuscript. Another major strength of the manuscript is the thorough experimental data and rationale that was presented makes a convincing case that supports most steps in the proposed mechanism. Specifically, the data convinced the reviewer that CD98hc is required for cell response to mechanical signals, CD98hc is upstream of RhoA activation, and RhoA activation isn’t defective; CD98hc is involved in cell contractility upstream of RhoA activation and its presence is required for the expression of stiffness-dependent genes; CD98hc was required for normal tissue elasticity, and normal production of 3 collagen-crosslinking enzymes; and CD98hc influences DES2 levels, and this sphingomyelin levels, and both normal sphingolipid levels and CD98hc are required for cell response to mechanical stimuli. Clearly a significant amount of careful planning, expertise, and time was required to complete the wide range of experiments that were used to demonstrate these aspects of the proposed mechanism.

The major weakness is that the experimental evidence more weakly supported the involvement of GEF-H1, and

especially cholesterol enriched membrane microdomains in the proposed mechanism. These are the major weaknesses, which are described below.

We deeply acknowledge the reviewer for his comments and insight, and particularly for his appreciation of our work. The reviewer raises two major points that we would like to discuss and that we addressed experimentally.

1. In regard to GEF-H1, the reviewer did not understand how the GST-RhoA pulldown showed GEF-H1 was not functioning properly in the CD98hc null cells, or how the membrane association/cell fractionation experiment showed the GEF-H1 was inactive in the CD98hc cells. Can the authors clarify their explanation of what their findings show and the rationale for how they lead to the conclusion that GEF-H1 was not functioning properly and inactive in the GEF-H1 cells?

The GST-RhoA-17A assay is designed to measure the amount of biochemically active RhoA-specific GEF at any given time in cell lysates thereby reflecting the activity of these GEFs in cells. This assay is based on the particular properties of the RhoA-17A mutant and the reactive intermediates occurring during the activation of RhoA by its GEF. Briefly, this mutant form of RhoA, coined nucleotide-free, mimics a reactive intermediate form of RhoA occurring during the process of its activation by a GEF. The activation of RhoA occurs through the exchange of a GDP molecule for a GTP molecule catalyzed by a GEF. This process requires the displacement of the RhoA-bound nucleotide by the GEFs which results in a short-lived RhoA-nucleotide free intermediate which has a high affinity for active GEFs (stably mimicked by the RhoA-17A mutant in the assay). The assay is therefore designed to use GST-RhoA-17A as a bait to trap active RhoA-specific GEF in a cell or tissue lysate. The output of the assay is therefore the identification and/or quantification of the amount of GEF trapped on GST-RhoA-17A beads. However, the limitation of the assay is that it measures the amount of active GEF only from a biochemical standpoint, ie the amount of GEF in an active conformation, without taking into consideration its subcellular localization which may be critical to translate this biochemical capacity to activate RhoA into a bona fide biological activation of RhoA. Here, in our manuscript, we measured the biochemical activation of the two GEFs that have been reported to activate RhoA in response to mechanical force application on integrins, namely GEF-H1 and LARG. We found, on the basis of this assay, that both GEFs were not activated properly in response to that stimulation in CD98hc null cells. This prompted us to assess how that defective activation occurred and we investigated the upstream activators of these GEFs which may be the simplest explanation. While this is easy to explain in the case of LARG since its upstream activators, Src kinases, are not properly activated either, this defective activation is more difficult to understand regarding GEF-H1 since its direct upstream activators, ERK1/2, seem to be activated properly in CD98hc null cells. As a note, the activation of GEF-H1 occurs via a direct phosphorylation by ERK1/2.

A classical paradigm regarding the regulation of Rho proteins is that their biochemical regulation is coupled to their subcellular localization and their cycling to the plasma membrane. We hypothesized that such a similar regulation may occur at the level of GEF-H1 which may justify why despite the activation of ERK1/2, GEF-H1 remained inactive, ie sequestered in a subcellular compartment preventing its activation by ERK1/2. We unsuccessfully tried to document ERK1/2-dependent GEF-H1 phosphorylation (data not shown). We assessed GEF-H1 localization in membrane fractions and observed that it was sequestered in membrane fractions in CD98hc null cells (Fig 6C). We then coupled cell fractionation to GST-RhoA-17A assay to assess GEF-H1 biochemical activation in membrane fractions (Fig 6D). We conclude from that experiment that most of GEF-H1 is sequestered in membrane fraction in CD98hc null cells in an inactive conformation. In light of the defect in

sphingolipid synthesis in these cells, we hypothesized that GEF-H1 may be sequestered in an inactive conformation in membranes and could be released from the membrane by shuttling first to lipid rafts/membrane microdomains in order to become available again for activation by ERK1/2. This was our rationale for looking at lipid rafts and no other aspects of sphingolipids signaling such as bioactive lipids. We assessed the localization of GEF-H1 in membrane microdomains by sucrose gradient purification, with the limitations highlighted by the reviewer. It turns out that GEF-H1 is not present in this membrane fraction in CD98hc null cells (Fig 6F). We also manipulated lipid rafts/membrane microdomains in control or CD98hc null cells to validate our hypothesis: we observe that cholesterol depletion in control cells results in sequestration of GEF-H1 in membrane fraction (Fig 6G) while addition of C12SM in CD98hc null cells, at least partially, releases GEF-H1 from membrane fraction (Fig 6H). Similarly, GEF-H1 was sequestered to membranes in C330S cells (Fig S6C).

2. The involvement of cholesterol enriched membrane microdomains seems to be based on the findings of how sphingolipid levels were correlated with cell response/RhoA activation to mechanical signals, the finding that CD98hc null cells have lower cholesterol levels, and the GEF-H1 from lysed cells is recovered in a membrane fraction when separated on a sucrose gradient. Indeed, the idea that the primary function of cholesterol and sphingolipids in cells is to form special membrane microdomains that recruit a subset of membrane proteins that form complexes and mediate function is a popular hypothesis, and membrane fractionation studies use to be a popular technique for “detecting” them. However, in the past 5 years, a few new complementary high resolution imaging methods have disproved the idea that cholesterol enriched microdomains are present in the membrane—see Eggling and coworkers in *Nature Comm* 2014, 5:5142, Kraft and coworkers in *J Biol Chem* 2013, 288:16855, and

Fujiwara and coworkers in *Traffic* 2014, 15:583. I do not doubt that CD98hc influences sphingolipid levels, and sphingolipids and possibly cholesterol are involved in CD98hc-mediated mechanical stimuli sensing. However, the data presented in this manuscript does not convince the reviewer that cholesterol enriched membrane microdomains are involved in this mechanism, especially in light of the aforementioned studies that indicate such domains do not exist. Furthermore, the cell lysis and sucrose density separation technique used in this manuscript is no longer considered a reliable method to assess a component's intracellular location. At best, it provides information about a component's affinity for membranous material, and the value of that information is questionable. It seems more likely that sphingolipids have some other role in this process. Several studies have shown that sphingolipids can selectively bind to distinct sites on membrane proteins, which modulates their conformation and activity (see Brugger and coworkers in *Nature* 2012, 481:525 and Barrantes and coworkers in *BBA* 2009, 1788:2345). Furthermore, sphingomyelin metabolites are signaling molecules, functioning as second messengers that modulate a variety of cell processes; Yusuf Hannun has published extensively on sphingolipid signaling mechanisms. Either type of mechanism is more plausible than one involving cholesterol enriched microdomains. It seems that the precise mechanism for why sphingomyelin is required for CD98hc-modulated mechanical sensing was not the author's main focus, and cholesterol enriched microdomains were suggested because they are the “go to” explanation for processes that involve cholesterol or sphingolipids. If that is the case, I suggest omitting mentioning cholesterol enriched membrane microdomains and simply say the exact mechanism for why sphingomyelin is required for CD98hc-modulated mechanical sensing will require future studies to elucidate.

(I do not think that the mechanism of sphingomyelin involvement needs to be elucidated to warrant publication of this manuscript.) Alternatively, it would be acceptable to mention the other possible mechanisms in addition to cholesterol enriched membrane microdomains to explain the sphingomyelin involvement. If the authors are

committed to insist the mechanism involves cholesterol enriched microdomains, the authors would need to simultaneously image GEF-H1 and cholesterol in cells and show they are colocalized in membrane microdomains in normal cells but not CD98hc null cells to provide more convincing support for that aspect of their mechanism.

We deeply acknowledge the reviewer for his constructive comments. As foreboded by the reviewer, we are not experts in sphingolipid biology and we welcome any suggestion. Indeed, we originally focused on cholesterol-enriched membrane microdomains (CEMM) on the basis of the data aforementioned by the reviewer. Among these, we would like to emphasize on the fact that cholesterol levels are reduced in CD98hc null cells and that manipulating solely cholesterol levels with cholesterol oxidase or β -methylcyclodextrin (not shown) does recapitulate the effects of CD98hc depletion on mechanosensing which from our point of view also pointed out at CEMM independently of sphingolipids and suggest that the same effect can be obtained without altering sphingolipids themselves but a structure containing both cholesterol and sphingolipids. Nevertheless, we were not fully aware that the existence of these microdomains was still so controversial and we acknowledge the reviewer for pointing that out and offering alternative possibilities. Among these other possibilities, we are aware that sphingolipids can behave as second messengers. Originally, we had dismissed that possibility on the basis of the sucrose gradient experiments and the effect of manipulating sphingolipids/cholesterol levels on GEF-H1 localization. Nevertheless, we fully agree that while we show that sphingolipids are involved in the regulation of mechanosensing via GEF-H1, we do not formally show it is solely dependent on CEMM and we did not exclude other possibilities. We would also like to mention that although we cannot formally exclude a second messenger function, the signal triggering RhoA activation is mechanical through integrins and has been precisely characterized at each step at the molecular level by us and others (Guilluy et al., 2011; Schiller et al., 2013). This would most likely imply that a constant stream of second messenger would prime cells in the control condition and would be missing in the CD98hc null cells. Given the general mode of action of second messengers, this is not a mechanism that we would expect.

We do not necessarily want to link mechanosensing and "lipid rafts", and we are totally open to alternative explanations including sphingolipids as second messengers. From reading the articles and reviews suggested by the reviewer, it stands out that many sphingolipids can behave as second messengers including ceramides, sphingosine and sphingosine-1-phosphate (S1P) making it particularly difficult to assess their involvement since we cannot reasonably test every candidate. S1P is a particularly interesting candidate since it can activate RhoA (Vouret-Craviari et al., 2002) and has been associated with the regulation of cell adhesion/spreading/migration. For those reasons, we assessed whether it could be an alternative mechanism accounting for sphingolipid regulation of mechanosensing. We treated cells with the sphingosine kinase inhibitor di-methyl sphingosine (DMS) prior to mechanical stimulation with magnetic beads. We found that DMS treatment had no effect on mechanical RhoA activation. This seems to exclude at least S1P as second messenger in this process. Also, as suggested by the reviewer, we have performed simultaneous staining of cholesterol using filipin III as a tracer and GEF-H1 using an anti-GEF-H1 antibody. The results were quite surprising to us but in light of all the data gathered in this manuscript regarding GEF-H1 localization, they make completely sense. We stained both cholesterol and GEF-H1 in control, CD98hc null and Des2 shRNA cells. We observe that cholesterol staining strongly decreases in CD98hc null cells and DES2 shRNA cells as compared to control cells (Fig S7). Indeed, it diminishes to a similar extent as control cells treated with cholesterol oxidase. We also observed that the cholesterol staining was more heterogenous in DES2 shRNA cells as compared to CD98hc null cells which, in our opinion, may be justified as DES2 depletion is not complete as compared to the genetic invalidation of CD98hc

and because not all cells were transfected to the same extent. Regarding GEF-H1 staining and its colocalization with cholesterol, we observed in CD98hc null cells that while cholesterol levels are decreased, GEF-H1 staining is mostly in the plasma membrane, however not colocalized with cholesterol. While initially surprising, we realized that we observed that GEF-H1 is constitutively recruited to the membrane in CD98hc null cells (Fig 7) which paralleled our observation by immunofluorescence. This also confirmed our hypothesis that GEF-H1 needs membrane microdomains to be released in the cytosol and justifies why we see that strong signal in the membrane of CD98hc null cells. Similar observations were made in DES2 shRNA cells with the heterogeneity aforementioned. In control cells, we could observe cells in which cholesterol and GEF-H1 were colocalized in the plasma membrane (specifically in lamellipodia in some instances) (Fig 7) but we also observed cells in which GEF-H1 was not localized to the plasma membrane (Reviewers Fig 4). Generally, GEF-H1 localization was less membranous as compared to CD98hc null cells. We were dismayed at first but soon realized that GEF-H1 cycling is a dynamic process and that we were observing snapshots of GEF-H1 cycling at different times in different cells depending also in the amount of cholesterol. Since GEF-H1 most likely requires shuttling to membrane microdomains to be released in the cytosol, its cycling was not blocked in control cells like in CD98hc cells and it was less likely to accumulate in a specific location such as the plasma membrane. Therefore, depending on cells, we could observe colocalization with cholesterol, but we could also observe either accumulation of GEF-H1 in the membrane of cells with less cholesterol (Fig 7) or cytosolic localization in cells where it had been released from the membrane because of the presence of membrane microdomains (reviewers fig 4). We think that altogether, these data indicate that cholesterol and GEF-H1 can colocalize, they also support our model of sequestration of GEF-H1 in the plasma membrane of CD98hc null cells and the requirement for membrane microdomain for its cytosolic release.

Also, after thoroughly searching the literature, we found evidence that CD98hc could localize in membrane microdomains together with several of the proteins involved in the regulation of the process described herein: namely Integrin b1, the molecular chaperone grp78 as well as other molecular chaperones of the hsp70/90 families and several members of the src family kinases including fyn, yes and lyn (supplemental figure S1 of (Schroeder et al., 2012)).

Based on all these considerations and in light of the advice from the reviewer, we propose to mention in the text the current main hypothesis of the involvement of membrane microdomains and the alternative possibilities mentioned by the reviewer. We tried to mention "membrane microdomain" instead of "CEMM" in the text since existence of CEMM is still so controversial. We tried to present both possibilities, indicate that we take them both into account by testing S1P for instance and that on the basis of the data available, we favor the microdomain possibility without excluding others at this stage. Therefore, we also included in the manuscript the experiment assessing the involvement of sphingosine-1-phosphate in Figure 7. Also, we included the cholesterol/GEF-H1 staining in Figure 7. In summary, we want to be very cautious here and we state in the text that we favor membrane microdomains on the basis of the data in hand but that we do not/cannot formally exclude other possibilities such as sphingolipids as second messenger for instance.

We hope the reviewer will find this statement reasonable and we are very much willing to discuss this further if required by the reviewer. We hope to give the reader a fair depiction of the situation in order to make up his/her own mind.

Reviewer #3, an expert in metabolomics (Remarks to the Author):

Comments on Nat-COM 152284

These comments are on the metabolomics results only.

There are several issues around the metabolomics work. The extraction used is not specific for sphingolipids.

There major results need to be confirmed by a method that has been validated for sphingolipid analysis.

It is unclear what the number of samples were that were used in the metabolomics screen. There is no information on the error of the measurement.

There are no details on the data processing of the metabolomics data. From my experience with Metabolon I expect that the method used yielded relative concentrations, that were log transformed and then scaled and the difference expressed as relative fold changes, but this needs to be clarified. With this approach information is lost about the actual difference in concentration between metabolites and over emphasises the difference between low concentration metabolites which measurements are less precise. Again by using a targeted method for sphingolipids it becomes only possible to really assess the effect on the sphingomyelin metabolism.

We acknowledge the reviewer on his comments on the metabolomics study and his insights as an expert in the field. As probably noticed by the reviewer, we are not experts on metabolomics and we largely relied on the expertise and advice provided by Metabolon on the experimental set up and the processing of the raw data. Clearly, as mentioned by the reviewer, the assay was in no way designed to specifically assess sphingolipids level but to assess the variations of a wide range of metabolites indeed. We shipped snap frozen cell pellets to Metabolon which performed metabolites extraction, identification and analysis (n=4 replicates shipped for each sample).

Here is the precise protocol as communicated to us by Metabolon:

Sample Accessioning: Following receipt, samples were inventoried and immediately stored at -80oC. Each sample received was accessioned into the Metabolon LIMS system and was assigned by the LIMS a unique identifier that was associated with the original source identifier only. This identifier was used to track all sample handling, tasks, results, etc. The samples (and all derived aliquots) were tracked by the LIMS system. All portions of any sample were automatically assigned their own unique identifiers by the LIMS when a new task was created; the relationship of these samples was also tracked. All samples were maintained at -80oC until processed.

Sample Preparation: Samples were prepared using the automated MicroLab STAR® system from Hamilton Company. Several recovery standards were added prior to the first step in the extraction process for QC purposes. To remove protein, dissociate small molecules bound to protein or trapped in the precipitated protein matrix, and to recover chemically diverse metabolites, proteins were precipitated with methanol under vigorous shaking for 2 min (Glen Mills GenoGrinder 2000) followed by centrifugation. The resulting extract was divided into five fractions: two for analysis by two separate reverse phase (RP)/UPLC-MS/MS methods with positive ion mode electrospray ionization (ESI), one for analysis by RP/UPLC-MS/MS with negative ion mode ESI, one for analysis by HILIC/UPLC-MS/MS with

negative ion mode ESI, and one sample was reserved for backup. Samples were placed briefly on a TurboVap® (Zymark) to remove the organic solvent. The sample extracts were stored overnight under nitrogen before preparation for analysis.

QA/QC: Several types of controls were analyzed in concert with the experimental samples: a pooled matrix sample generated by taking a small volume of each experimental sample (or alternatively, use of a pool of well-characterized human plasma) served as a technical replicate throughout the data set; extracted water samples served as process blanks; and a cocktail of QC standards that were carefully chosen not to interfere with the measurement of endogenous compounds were spiked into every analyzed sample, allowed instrument performance monitoring and aided chromatographic alignment. Tables 1 and 2 describe these QC samples and standards. Instrument variability was determined by calculating the median relative standard deviation (RSD) for the standards that were added to each sample prior to injection into the mass spectrometers. Overall process variability was determined by calculating the median RSD for all endogenous metabolites (i.e., non-instrument standards) present in 100% of the pooled matrix samples. Experimental samples were randomized across the platform run with QC samples spaced evenly among the injections.

Ultrahigh Performance Liquid Chromatography-Tandem Mass Spectroscopy (UPLC-MS/MS): All methods utilized a Waters ACQUITY ultra-performance liquid chromatography (UPLC) and a Thermo Scientific Q-Exactive high resolution/accurate mass spectrometer interfaced with a heated electrospray ionization (HESI-II) source and Orbitrap mass analyzer operated at 35,000 mass resolution. The sample extract was dried then reconstituted in solvents compatible to each of the four methods. Each reconstitution solvent contained a series of standards at fixed concentrations to ensure injection and chromatographic consistency. One aliquot was analyzed using acidic positive ion conditions, chromatographically optimized for more hydrophilic compounds. In this method, the extract was gradient eluted from a C18 column (Waters UPLC BEH C18-2.1x100 mm, 1.7 μ m) using water and methanol, containing 0.05% perfluoropentanoic acid (PFPA) and 0.1% formic acid (FA). Another aliquot was also analyzed using acidic positive ion conditions, however it was chromatographically optimized for more hydrophobic compounds. In this method, the extract was gradient eluted from the same afore mentioned C18 column using methanol, acetonitrile, water, 0.05% PFPA and 0.01% FA and was operated at an overall higher organic content. Another aliquot was analyzed using basic negative ion optimized conditions using a separate dedicated C18 column. The basic extracts were gradient eluted from the column using methanol and water, however with 6.5mM Ammonium Bicarbonate at pH 8. The fourth aliquot was analyzed via negative ionization following elution from a HILIC column (Waters UPLC BEH Amide 2.1x150 mm, 1.7 μ m) using a gradient consisting of water and acetonitrile with 10mM Ammonium Formate, pH 10.8. The MS analysis alternated between MS and data-dependent MS_n scans using dynamic exclusion. The scan range varied slightly between methods but covered 70-1000 m/z. Raw data files are archived and extracted as described below.

Bioinformatics: The informatics system consisted of four major components, the Laboratory Information Management System (LIMS), the data extraction and peak-identification software, data processing tools for QC and compound identification, and a collection of information interpretation and visualization tools

for use by data analysts. The hardware and software foundations for these informatics components were the LAN backbone, and a database server running Oracle 10.2.0.1 Enterprise Edition.

LIMS: The purpose of the Metabolon LIMS system was to enable fully auditable laboratory automation through a secure, easy to use, and highly specialized system. The scope of the Metabolon LIMS system encompasses sample accessioning, sample preparation and instrumental analysis and reporting and advanced data analysis. All of the subsequent software systems are grounded in the LIMS data structures. It has been modified to leverage and interface with the in-house information extraction and data visualization systems, as well as third party instrumentation and data analysis software.

Data Extraction and Compound Identification: Raw data was extracted, peak-identified and QC processed using Metabolon's hardware and software. These systems are built on a web-service platform utilizing Microsoft's .NET technologies, which run on high-performance application servers and fiber-channel storage arrays in clusters to provide active failover and load-balancing. Compounds were identified by comparison to library entries of purified standards or recurrent unknown entities. Metabolon maintains a library based on authenticated standards that contains the retention time/index (RI), mass to charge ratio (m/z), and chromatographic data (including MS/MS spectral data) on all molecules present in the library. Furthermore, biochemical identifications are based on three criteria: retention index within a narrow RI window of the proposed identification, accurate mass match to the library +/- 10 ppm, and the MS/MS forward and reverse scores between the experimental data and authentic standards. The MS/MS scores are based on a comparison of the ions present in the experimental spectrum to the ions present in the library spectrum. While there may be similarities between these molecules based on one of these factors, the use of all three data points can be utilized to distinguish and differentiate biochemicals. More than 3300 commercially available purified standard compounds have been acquired and registered into LIMS for analysis on all platforms for determination of their analytical characteristics. Additional mass spectral entries have been created for structurally unnamed biochemicals, which have been identified by virtue of their recurrent nature (both chromatographic and mass spectral). These compounds have the potential to be identified by future acquisition of a matching purified standard or by classical structural analysis.

Curation: A variety of curation procedures were carried out to ensure that a high-quality data set was made available for statistical analysis and data interpretation. The QC and curation processes were designed to ensure accurate and consistent identification of true chemical entities, and to remove those representing system artifacts, mis-assignments, and background noise. Metabolon data analysts use proprietary visualization and interpretation software to confirm the consistency of peak identification among the various samples. Library matches for each compound were checked for each sample and corrected if necessary.

Metabolite Quantification and Data Normalization: Peaks were quantified using area-under-the-curve. For studies spanning multiple days, a data normalization step was performed to correct variation resulting from instrument inter-day tuning differences. Essentially, each compound was corrected in run-day blocks by registering the medians to equal one (1.00) and normalizing each data point proportionately (termed the "block correction"; Figure 2). For studies that did not require more than one

day of analysis, no normalization is necessary, other than for purposes of data visualization. In certain instances, biochemical data may have been normalized to an additional factor (e.g., cell counts, total protein as determined by Bradford assay, osmolality, etc.) to account for differences in metabolite levels due to differences in the amount of material present in each sample.

We did not disclose all experimental details in the method section but for clarity we would be willing to include this information in the method section upon request by the reviewer or editor.

From our understanding of the processing and analysis of the metabolomics screen, we agree that the assay was not designed to provide absolute quantification of metabolites content but rather to detect variations of metabolites quantity as compared to a reference control sample. The goal was to attempt to identify metabolites and a metabolic pathway that could account for the defect in mechanosensing and would be affected along with defective mechanosensing. Therefore, the assay was designed to be as unbiased as possible and the experimental conditions were designed to include as many different metabolites as possible. We compared various CD98hc mutant expression conditions to a control condition, respectively control CD98hc fl/fl cells for CD98hc null cells or Fsp1-CRE CD98fl/fl wt CD98hc re-expressing cells for the mutant conditions. We obtained fold variations of metabolites content over control which pointed out at sphingolipid metabolism. This led us to focus of sphingolipid metabolism and sphingolipids level. The implication of this pathway in the regulation of mechanosensing was then assessed by manipulating sphingolipids and cholesterol levels while monitoring cellular response to mechanical stimulation (Figs 4F, 4G, 5J, 6G, 6H, S4E and S4F). Additionally, we have now quantified sphingolipid content using a sphingomyelin assay kit which points at a global reduction of roughly 25% when comparing CD98hc null cells to control cells, very much in the range of reduction observed for single sphingomyelins (Table 2). This was added to Fig S4.

Minor comments

Metabolon's measurements have a relative standard deviation between 5 and 20%. This means that the numbers given in supplementary table 2 are not presented correctly. With a 10% RSD that means for ketosphinganine the first number of 0.3196 should be shown as 0.3 because the further digits can not be measured. The numbers in the table suggest a level of precision that is not there.

Again, we acknowledge the reviewer for his expertise and his advice regarding the interpretation of the metabolomics data. Here again, the numbers were reported as communicated to us by Metabolon. As expected by the reviewer, Metabolon reported an RSD of 8%. We are not sure to fully understand the reviewer's calculation but, as an expert in the field we are very much willing to trust him/her. We modified table 2 according to the reviewer's comments. Alternatively, we can also provide, as a table the raw data and calculated values as provided by Metabolon so that expert readers can make up their own mind on the data.

References

Achterberg, V.F., Buscemi, L., Diekmann, H., Smith-Clerc, J., Schwengler, H., Meister, J.-J.,

Wenck, H., Gallinat, S., and Hinz, B. (2014). The Nano-Scale Mechanical Properties of the Extracellular Matrix Regulate Dermal Fibroblast Function. *J. Invest. Dermatol.* *134*, 1862–1872.

Aghajanian, A., Wittchen, E.S., Campbell, S.L., and Burridge, K. (2009). Direct Activation of RhoA by Reactive Oxygen Species Requires a Redox-Sensitive Motif. *PLoS ONE* *4*, e8045.

Arthur, W.T., and Burridge, K. (2001). RhoA inactivation by p190RhoGAP regulates cell spreading and migration by promoting membrane protrusion and polarity. *Mol. Biol. Cell* *12*, 2711–2720.

Arthur, W.T., Petch, L.A., and Burridge, K. (2000). Integrin engagement suppresses RhoA activity via a c-Src-dependent mechanism. *Curr. Biol.* *10*, 719–722.

Boulter, E., Garcia-Mata, R., Guilluy, C., Dubash, A., Rossi, G., Brennwald, P.J., and Burridge, K. (2010). Regulation of Rho GTPase crosstalk, degradation and activity by RhoGDI1. *Nat Cell Biol* *12*, 477–483.

Boulter, E., Estrach, S., Errante, A., Pons, C., Cailleteau, L., Tissot, F., Meneguzzi, G., and Feral, C.C. (2013). CD98hc (SLC3A2) regulation of skin homeostasis wanes with age. *J Exp Med* *210*, 173–190.

Cormerais, Y., Giuliano, S., LeFloch, R., Front, B., Durivault, J., Tambutte, E., Massard, P.-A., de la Ballina, L.R., Endou, H., Wempe, M.F., et al. (2016). Genetic Disruption of the Multifunctional CD98/LAT1 Complex Demonstrates the Key Role of Essential Amino Acid Transport in the Control of mTORC1 and Tumor Growth. *Cancer Res.* *76*, 4481–4492.

Estévez, R., Camps, M., Rojas, A.M., Testar, X., Devés, R., Hediger, M.A., Zorzano, A., and Palacín, M. (1998). The amino acid transport system y⁺ L/4F2hc is a heteromultimeric complex. *FASEB J.* *12*, 1319–1329.

Fabrias, G., Muñoz-Olaya, J., Cingolani, F., Signorelli, P., Casas, J., Gagliostro, V., and Ghidoni, R. (2012). Dihydroceramide desaturase and dihydrosphingolipids: Debutant players in the sphingolipid arena. *Prog. Lipid Res.* *51*, 82–94.

Fenczik, C.A., Sethi, T., Ramos, J.W., Hughes, P.E., and Ginsberg, M.H. (1997). Complementation of dominant suppression implicates CD98 in integrin activation. *Nature* *390*, 81–85.

Feral, C.C., Nishiya, N., Fenczik, C.A., Stuhlmann, H., Slepak, M., and Ginsberg, M.H. (2005). CD98hc (SLC3A2) mediates integrin signaling. *Proc Natl Acad Sci U S A* *102*, 355–360.

Feral, C.C., Zijlstra, A., Tkachenko, E., Prager, G., Gardel, M.L., Slepak, M., and Ginsberg, M.H. (2007). CD98hc (SLC3A2) participates in fibronectin matrix assembly by mediating integrin signaling. *J Cell Biol* *178*, 701–711.

Guilluy, C., Swaminathan, V., Garcia-Mata, R., O'Brien, E.T., Superfine, R., and Burridge, K. (2011). The Rho GEFs LARG and GEF-H1 regulate the mechanical response to force on integrins. *Nat Cell Biol* *13*, 722–727.

Nicklin, P., Bergman, P., Zhang, B., Triantafellow, E., Wang, H., Nyfeler, B., Yang, H., Hild, M., Kung, C., Wilson, C., et al. (2009). Bidirectional Transport of Amino Acids Regulates mTOR and Autophagy. *Cell* *136*, 521–534.

Schiller, H.B., Hermann, M.-R., Polleux, J., Vignaud, T., Zanivan, S., Friedel, C.C., Sun, Z., Raducanu, A., Gottschalk, K.-E., Thery, M., et al. (2013). β 1- and β v-class integrins cooperate to regulate myosin II during rigidity sensing of fibronectin-based microenvironments. *Nat. Cell Biol.* *15*, 625–636.

Schroeder, N., Chung, C.-S., Chen, C.-H., Liao, C.-L., and Chang, W. (2012). The Lipid Raft-Associated Protein CD98 Is Required for Vaccinia Virus Endocytosis. *J. Virol.* *86*, 4868–4882.

Vouret-Craviari, V., Bourcier, C., Boulter, E., and van Obberghen-Schilling, E. (2002). Distinct

signals via Rho GTPases and Src drive shape changes by thrombin and sphingosine-1-phosphate in endothelial cells. *J Cell Sci* *115*, 2475–2484.

Reviewer Figure 1 – Cell spreading is not affected by RhoA depletion. **A** Control or RhoA miR shRNA expressing HeLa cells were plated on FN-coated glass coverslips for indicated times prior to fixation and actin staining (phalloidin). **B** Schematic depiction of the design of the GFP RhoA miR shRNA construct. **C** Control or RhoA miR shRNA expressing HeLa cells were lysed and RhoA expression was assessed by Western blotting.

Reviewer Figure 2 - **A** Depletion of cholesterol alters both RhoA localization and activation. Dermal fibroblasts were treated with cholesterol oxidase for 2 hours prior to membrane purification or GST-RBD pull-down assay. **B** RhoA does not localize to membranes in CD98hc null cells. Control or CD98hc null cells were incubated with FN-coated magnetic beads prior to mechanical stimulation and membrane fractionation.

Reviewer Figure 3

- 1 - Carnitine Metabolism
- 2 - Fatty Acid Metabolism
- 3 - Fatty Acid, Amino
- 4 - Phosphatidylserine
- 5 - Inositol Metabolism
- 6 - Fatty Acid, Branched
- 7 - Neurotransmitter
- 8 - Mevalonate Metabolism

- 9 - Lysoplasmalogen
- 10 - Glycerolipid Metabolism
- 11 - Fatty Acid, Keto
- 12 - Medium Chain Fatty Acid
- 13 - Ketone Bodies
- 14 - Fatty Acid Metabolism (Acyl Choline)
- 15 - Secondary Bile Acid Metabolism
- 16 - Fatty Acid Metabolism (Acyl Glycine)
- 17 - Fatty Acid Synthesis

Reviewer Figure 4

wt/kO du

Key	Name	Parameter	Gate
---	22/05/12.003	FL2-H	G1
---	22/05/12.012	FL2-H	G1
---	22/05/12.006	FL2-H	G1
---	22/05/12.019	FL2-H	G1

Reviewer Figure 4 - Scan of the cell surface expression of integrin av in control (wt) and CD98hc null (KO) cells

Reviewers' comments:

Reviewer #1 (Remarks to the Author):

The major concern was the lack of mechanistic insight linking integrin as mechanoreceptors important for mechanosensing and CD98/DSE2. The request was to identify the potential role of CD98 as a chaperone or as an actor in integrin dynamics (integrin sequestration, integrin internalization, etc..) due to the defect of sphingolipid metabolism and in turn an expected defect in membrane composition.

The authors show that CD98 impacts sphingolipid synthesis through the folding and the stabilization of delta-4-desaturase DES2. In turn, defect of sphingolipid synthesis prevents proper membrane recruitment, shuttling and activation of upstream regulator of RhoA including Src kinase and GEFH1 which impact mechanosensing. The authors have also shown that CD98 is required for the Hsp-dependent folding and stabilization of DES2. They have shown that the loss of CD98 induces the constitutive association between DES2 and grp94 and have tested the consequences of DES2 silencing. DES2 does not act as a chaperone but is involved in intervention of molecular chaperone.

The role of CD98 in sphingolipid metabolism cannot be restricted to a defect of GEF-H1 shuttling or recruitment. It is associated with a change in membrane composition and organization meaning that a change in integrin dynamics is expected. Integrin dynamics (integrin motility) cannot be excluded from this study and should provide the real explanation for mechanosensing alteration and GEF-H1 recruitment/shuttling defect.

The loss of DES2 does not recapitulate exactly the loss of CD98. Fig. 6F shows that DES2 depleted cells generated more force as contrary to CD98 null cells. As said by the authors, DES2 might contribute to others mechanical processes. One possibility is that other mechanical processes are due to a change of integrin motility (even though the level of integrin expression and activation are not changed). The authors have also noticed the decrease of flotillin, cholesterol and sphingomyelin level, another proof supporting a change in membrane composition and in nanodomain organization.

The authors checked that sphingolipid metabolism do not change the level of integrin expression or integrin activation but did not look at integrin dynamics. Sphingolipids are essential constituents of the plasma membrane and play an important role in signal transduction by modulating clustering and dynamics of membrane receptors. If the approach identifying the defect of sphingolipid metabolism is accepted (by referee 3), excluding the role of sphingolipids in membrane composition and consequently in integrin dynamics is not acceptable. DES2 is an ER associated protein. Some results support a model in which ER-PM contact sites provide a nexus for coordinating the complex interrelationship between sterols, sphingolipids, and phospholipids that maintain PM composition and integrity (Quon E, Sere YY, Chauhan N, et al PLoS Biol. 2018). Moreover another publication has shown that changes in membrane sphingolipid composition modulate dynamics and adhesion of integrin nanoclusters. More precisely, single molecule dynamic approaches have demonstrated that the local lipid environment regulates adhesion of integrin receptors by impacting on their lateral mobility (Eich et al Sci Rep. 2016). Mechanosensing also depends on integrin motility meaning that integrin dynamics needs to be monitored with and without CD98 (FRAP, video for half-time, internalization) to see whether CD98 is affecting integrin dynamics through a defect of sphingolipid metabolism. No experiments in the revised version allow saying that the defect of mechanosensing occurs independently of direct integrin regulation. We can expect a defect of Rho activation in response to a change in integrin dynamics.

In other terms, the effect of CD98 loss on RhoA activation and GEF-H1 sequestration is convincing. However rho activation can result from a defect of integrin dynamics likely due to a defect of

sphingolipid metabolism and a change of membrane composition. The story linking sphingolipid metabolism and mechanosensing is restricted to GEF-H1 sequestration. The impact of sphingolipid metabolism on integrin dynamics cannot be excluded because a defect of integrin motility can act on rho activation and GEF-H1 sequestration. The effect of CD98 loss on integrin dynamics is missing. The interpretation of the results needs to be reconsidered.

Images of Immunofluorescence have to be improved in quality (Supplementary Figure 2).

The data showing colocalisation between cholesterol and GEF-H1 are not convincing (Fig.7G).

Reviewer #2 (Remarks to the Author):

This manuscript shows the amino acid transporter CD98hc is required for functional mechanosensing. Their results show that depletion of functional CD98hc impedes RhoA activation in response to mechanical signals. They also discovered that loss of functional CD98hc is correlated with a decrease in DES2, a desaturase enzyme that is required for sphingolipid biosynthesis, and consequently, also a decrease in cellular sphingolipid levels. By performing a large number of diverse experiments, they show that CD98hc seems to impair the proper folding of DES2, leading to a decrease in sphingolipid levels. Interestingly, metabolic screens revealed loss of CD98hc is accompanied by a decrease in cellular cholesterol. Imaging experiments showed that GEF-H1 seems to be stuck in the plasma membrane when sphingolipid and cholesterol levels are low, which impedes GEF-H1 cycling between the plasma membrane and cytosol that is required to activate RhoA in response to mechanical signals.

Overall, the cross-regulatory mechanism described in this manuscript is very interesting and novel, and I expect it will be of interest to the Nature Communications readership. The authors present a large amount of experimental data to support the proposed mechanism. They clearly state which aspects of the proposed mechanism need more investigation, and possible alternative mechanisms when appropriate. I find this more than sufficient as support for the conclusions drawn. The authors have adequately responded to my comments, and I have no further questions or suggestions for improvement.

Reviewer #3 (Remarks to the Author):

Comments on Nat-COM 152284

These comments are on the metabolomics results only.

I am very disappointed with the response of the authors. The authors have sent the samples to Metabolon and received the results back but have no ability to assess these results critically, neither did they aspire to do so. The only thing they have done is provide some blur from Metabolon about the method, but the authors do not know what it means. There is no validation of the results obtained and no ability to do so. This lack of critical thought and lack of understanding of what the results mean in this manuscript does not provide me with any confidence about the conclusions. Without the validation of the key results by a dedicated method, there is insufficient evidence. At the moment the authors do not know if the results of Metabolon are the result of changes in extraction efficiency due to other changes in the composition of the samples, or that there are genuine changes in sphingolipid metabolism (or both). I would, therefore, suggest to either reject this manuscript or ask the authors to team up with a metabolomics specialist to validate the results.

Point by point response to reviewer's comments – NCOMMS-17-33042B

Reviewer #1 (Remarks to the Author):

The major concern was the lack of mechanistic insight linking integrin as mechanoreceptors important for mechanosensing and CD98/DES2. The request was to identify the potential role of CD98 as a chaperone or as an actor in integrin dynamics (integrin sequestration, integrin internalization, etc..) due to the defect of sphingolipid metabolism and in turn an expected defect in membrane composition.

The authors show that CD98 impacts sphingolipid synthesis through the folding and the stabilization of delta-4-desaturase DES2. In turn, defect of sphingolipid synthesis prevents proper membrane recruitment, shuttling and activation of upstream regulator of RhoA including Src kinase and GEFH1 which impact mechanosensing. The authors have also shown that CD98 is required for the Hsp-dependent folding and stabilization of DES2. They have shown that the loss of CD98 induces the constitutive association between DES2 and grp94 and have tested the consequences of DES2 silencing. DES2 does not act as a chaperone but is involved in intervention of molecular chaperone.

The role of CD98 in sphingolipid metabolism cannot be restricted to a defect of GEF-H1 shuttling or recruitment. It is associated with a change in membrane composition and organization meaning that a change in integrin dynamics is expected. Integrin dynamics (integrin motility) cannot be excluded from this study and should provide the real explanation for mechanosensing alteration and GEF-H1 recruitment/shuttling defect.

The loss of DES2 does not recapitulate exactly the loss of CD98. Fig. 6F shows that DES2 depleted cells generated more force as contrary to CD98 null cells. As said by the authors, DES2 might contribute to others mechanical processes. One possibility is that other mechanical processes are due to a change of integrin motility (even though the level of integrin expression and activation are not changed). The authors have also noticed the decrease of flotillin, cholesterol and sphingomyelin level, another proof supporting a change in membrane composition and in nanodomain organization.

The authors checked that sphingolipid metabolism do not change the level of integrin expression or integrin activation but did not look at integrin dynamics. Sphingolipids are essential constituents of the plasma membrane and play an important role in signal transduction by modulating clustering and dynamics of membrane receptors. If the approach identifying the defect of sphingolipid metabolism is accepted (by referee 3), excluding the role of sphingolipids in membrane composition and consequently in integrin dynamics is not acceptable. DES2 is an ER associated protein. Some results support a model in which ER-PM contact sites provide a nexus for coordinating the complex interrelationship between sterols, sphingolipids, and phospholipids that maintain PM composition and integrity (Quon E, Sere YY, Chauhan N, et al PLoS Biol. 2018). Moreover another publication has shown that changes in membrane sphingolipid composition modulate dynamics and adhesion of integrin nanoclusters. More precisely, single molecule dynamic approaches have demonstrated that the local lipid environment regulates adhesion of integrin receptors by impacting on their lateral mobility (Eich et al Sci Rep. 2016). Mechanosensing also depends on integrin motility meaning that integrin dynamics needs to be monitored with and without CD98 (FRAP, video for half-time, internalization) to see whether CD98 is affecting integrin dynamics through a defect of sphingolipid metabolism. No experiments in the revised version allow saying that the defect of mechanosensing occurs independently of direct integrin regulation. We can expect a defect of Rho activation in response to a change in integrin dynamics.

In other terms, the effect of CD98 loss on RhoA activation and GEF-H1 sequestration is convincing. However rho activation can result from a defect of integrin dynamics likely due to a defect of sphingolipid metabolism and a change of membrane composition. The story linking sphingolipid metabolism and mechanosensing is restricted to GEF-H1 sequestration. The impact of sphingolipid metabolism on integrin dynamics cannot be excluded because a defect of integrin motility can act on rho activation and GEF-H1 sequestration. The effect of CD98 loss on integrin dynamics is missing. The interpretation of the results needs to be reconsidered.

We acknowledge the reviewer for his comments and we agree that, although integrin expression is not affected by loss of CD98hc, integrin dynamics could be affected by loss of CD98hc and sphingolipids which may result in an abnormal regulation of GEF-H1, SFK and mechanosensing. We performed multiple experiments to address that matter including integrin dynamics and integrin trafficking as suggested by the reviewer.

First, we assessed integrin dynamics at the plasma membrane by FRAP. Briefly, we expressed GFP-tagged integrin $\alpha 5$ or integrin $\beta 3$ subunits in control (WT), CD98hc null cells or C330S cells in order to monitor the dynamics of $\alpha 5\beta 1$ and $\alpha \nu\beta 3$ respectively. The results are presented in Figure 7 and Supplementary Figure 8. Briefly, for both integrins, we performed FRAP on fluorescent adhesions by bleaching the fluorophore with a cycle of 10 iterations

of illumination at 100% of laser's power to reach at least 90% of bleach depth. As controls, we monitored the background fluorescence by measuring the fluorescence of a dark area of similar size in the field of view and we also controlled for laser illumination variations and overall bleaching by monitoring the fluorescence level of an unbleached adhesion. The data obtained were processed in easy-FRAP web and normalized. From these normalized data, we performed curve fitting analysis in Matlab, as described in the Method section, and generated fit functions to calculate the t half (time required to reach half maximal mobile fraction) and mobile fraction. Normalized data are presented in Supp Fig. 8A and B, and plots of fit functions are presented in Supp Fig. 8C and D (with R^2 in figure legend to evaluate fitness of fit). From these fits, we could calculate the t half which is characteristic of the diffusion rate, and the mobile fraction. We observed that the differences in t half between cells were not statistically significant. In contrast, we did observe significant differences in the size of the mobile fractions. For both integrins, the mobile fraction was significantly reduced upon loss of CD98hc or expression of C330S mutant.

Second, we monitored integrin trafficking by performing a pulse-chase experiment. We tried to monitor both integrin β 1 and β 3, unfortunately, the β 3 antibody did not seem to stain integrin β 3 and adhesion complexes, therefore we focused solely on integrin β 1. Briefly, we labeled integrin β 1 using Alexa 488-labeled monoclonal antibody MB1.2 at 4°C for 20 minutes. This antibody labels integrin β 1 in adhesion complexes as shown below on figure 1.

Figure 1 – Immunofluorescence staining of integrin β 1 and β 3 with the antibodies used in this study

We then chased integrin internalization and recycling at 37°C for the indicated times. Cells were fixed, and we got rid of the extracellular staining that remained (very strong focal adhesion staining) and prevented from proper intracellular observation by using an anti-Alexa488 antibody to quench the fluorophore. These results are presented in Figure 7I. We observed that integrin β 1 was internalized in control cells as early as 30 minutes and we could even observe some recycling to the membrane and adhesion complexes after 60 minutes (Supp. Fig. 8F). In contrast, in CD98hc null cells, integrin β 1 was hardly detected in the cytosol after 30 or 60 minutes. The situation was intermediate in C330S cells with less recycling than in control cells but more than in CD98hc null cells.

Altogether, these results seem to indicate that integrins are immobilized at the membrane upon loss of CD98hc or expression of C330S. This parallels the constitutive recruitment of GEF-H1 at the plasma membrane and suggests that integrin, at least in part, may participate to this process as initially suggested by the reviewer. The text of the manuscript has been modified accordingly to present these results and mention the possibility that sphingolipids may regulate GEF-H1 localization by way of integrins as well as CEMM. We acknowledge the reviewer for this very insightful suggestion which expands our understanding of the process.

Changes made in the text, have been highlighted in blue.

Images of Immunofluorescence have to be improved in quality (Supplementary Figure 2).

The data showing colocalisation between cholesterol and GEF-H1 are not convincing (Fig.7G).

We acknowledge the reviewer for his careful assessment of the figures particularly immunofluorescence staining. As advised, we had a second look on the immunofluorescence images. After careful review, we believe the

criticisms of the reviewer may stem from the poor quality of the images which were not as sharp both in terms of resolution and color as the originals and from the size of the images in the figures. We believe that original images displayed a much better quality and that sharpness and colors were lost along the process of conversion of the files into pdf files. Therefore, we agree with the reviewer that under these conditions it was difficult to stand by our conclusions on the basis of these figures. We apologize to the reviewer and to the editor for this poor handling of the images which could have been avoided. In consequence, we modified the figures as detailed below (Figure 2) and we generated PDF files of superior quality to protect the quality of images from being degraded. As an example, the reviewer can now compare the quality in the following figure and by assessing the figures now provided during the submission process.

Figure 2 – Comparison of the immunofluorescence image quality between initial (upper panel) and current (lower panel) submission

Figure 7 has been split into two figures (new figures 7 and 8) and cholesterol immunofluorescence staining is now in the new figure 8. We changed the colors of the figure in order for the cholesterol staining to stand out: cholesterol staining which was acquired in grey levels and changed into blue to match the fluorophore, is now displayed in grey levels which is much more contrasted than the blue color used in the previous version. Also, the merge panel has been removed to be able to double the size of the pictures. The same color correction has been performed on the cropped images. We have also added arrows to point at the colocalization, these arrows are white or red respectively for GEF-H1 or cholesterol to clearly stand out from the staining. The reader may notice that the cholesterol staining in CD98hc null cells is extremely faint, this is a reality that does not stem from image manipulation. Cholesterol levels are reduced in these cells and staining was very faint on the coverslips.

In supplementary Figure 2, the YAP staining which has been quantified, has not been modified but the quality of the PDF file should have been improved since image compression was suppressed.

Reviewer #2 (Remarks to the Author):

This manuscript shows the amino acid transporter CD98hc is required for functional mechanosensing. Their results show that depletion of functional CD98hc impedes RhoA activation in response to mechanical signals. They also discovered that loss of functional CD98hc is correlated with a decrease in DES2, a desaturase enzyme that is required for sphingolipid biosynthesis, and consequently, also a decrease in cellular sphingolipid levels. By performing a large number of diverse experiments, they show that CD98hc seems to impair the proper folding of DES2, leading to a decrease in sphingolipid levels. Interestingly, metabolic screens revealed loss of CD98hc is accompanied by a decrease in cellular cholesterol. Imaging experiments showed that GEF-H1 seems to be stuck in the plasma membrane when sphingolipid and cholesterol levels are low, which impedes GEF-H1 cycling between the plasma membrane and cytosol that is required to activate RhoA in response to mechanical signals.

Overall, the cross-regulatory mechanism described in this manuscript is very interesting and novel, and I expect it will be of interest to the Nature Communications readership. The authors present a large amount of experimental data to support the proposed mechanism. They clearly state which aspects of the proposed mechanism need more investigation, and possible alternative mechanisms when appropriate. I find this more than sufficient as support for the conclusions drawn. The authors have adequately responded to my comments, and I have no further questions or suggestions for improvement.

We acknowledge the reviewer for his insightful comments which helped improve the manuscript and consider alternative explanations.

Reviewer #3 (Remarks to the Author):

Comments on Nat-COM 152284

These comments are on the metabolomics results only.

I am very disappointed with the response of the authors. The authors have sent the samples to Metabolon and received the results back but have no ability to assess these results critically, neither did they aspire to do so. The only thing they have done is provide some blur from Metabolon about the method, but the authors do not know what it means. There is no validation of the results obtained and no ability to do so. This lack of critical thought and lack of understanding of what the results mean in this manuscript does not provide me with any confidence about the conclusions. Without the validation of the key results by a dedicated method, there is insufficient evidence. At the moment the authors do not know if the results of Metabolon are the result of changes in extraction efficiency due to other changes in the composition of the samples, or that there are genuine changes in sphingolipid metabolism (or both). I would, therefore, suggest to either reject this manuscript or ask the authors to team up with a metabolomics specialist to validate the results.

We apologize to the reviewer as we obviously misunderstood his/her previous comments and request. We were in no way meaning to be disrespectful or to disregard his/her comments. We apologize if such misunderstanding occurred.

We originally understood that the reviewer was questioning the experimental and technical approach as well as a lack of information regarding these. As previously indicated, we are no experts in metabolomics and we never claim to be such. Therefore, we originally hired Metabolon as experimental and analysis experts in order to carry out the experimental strategy we had devised.

As suggested by the reviewer, we have now sought for additional advice and guidance from metabolomics experts and we have teamed up with metabolomics and lipidomics expert Anne-Claude Gavin (group leader at EMBL, Heidelberg and EMBO member) and mass spectrometry expert Marco Hennrich from EMBL, Heidelberg. They have educated us on metabolomics data processing and statistical analysis, and together, we have critically assessed the metabolomics data. We have also provided them with access to all experimental details and QC checks from Metabolon.

First, regarding the assessment of data quality and various quality checks performed by Metabolon and double-checked by our metabolomics experts: Metabolon runs a number of internal and samples quality controls. As indicated by Metabolon, several types of controls were analyzed in concert with the experimental samples: a pooled matrix sample generated by taking a small volume of each experimental sample (or alternatively, use of a pool of well-characterized human plasma) served as a technical replicate throughout the data set; extracted water samples served as process blanks; and a cocktail of QC standards that were carefully chosen not to interfere with the measurement of endogenous compounds were spiked into every analyzed sample, allowed instrument performance monitoring and aided chromatographic alignment.

They determined instrument variability by calculating the median RSD from internal standards added to each sample prior to injection to the mass spectrometer. This RSD was determined to be 4%. They also determined process variability from the technical replicates which was calculated at a median RSD of 8%. In addition, we calculated the standard deviation and coefficient of variation of each metabolite over the four replicates that were analyzed for each cell type. This information is now included in the method section as well as detailed QC and sample processing explanation.

Regarding the data, the measurements of the individual samples were very reproducible, as visualized in the following plot.

Figure 3 – Distribution of the raw intensities of the metabolomics data before normalization (The central line in the box plots indicates the median, the bottom and top edges of the box the interquartile range (IQR) and the box plot whiskers represent 1.5 times the IQR)

Even though the reproducibility is quite good, systematic slight differences can have a huge impact on the results. We thus included a normalization step to correct for this. Also, as already mentioned by the reviewer, standardizing of raw data can lead to artificial changes especially of low abundant compounds. In our opinion some of the previously reported metabolites were close to the estimated noise level. We thus included a filtering step to remove these least abundant compounds. The normalized and filtered dataset was the basis to calculate p-values (t-test) and fold changes between the CD98hc re-expressing CD98hc knockout cells and all other conditions.

Figure 4 - Distribution of the raw intensities of the metabolomics data after normalization.

We then calculated basic statistics for each metabolite like standard deviation and coefficient of variation on this dataset which is now appended as a Supplementary Data File to the manuscript.

The conclusions of this analysis are that, in a very similar fashion as the original Metabolon analysis, the sphingolipid biosynthesis pathway is affected by loss of CD98hc or expression of C330S mutant (Fig. 4 and Supp Fig. 4). Regarding the latter, a detailed analysis of the sphingomyelin synthesis pathway revealed that 3-ketosphinganine, sphinganine and dehydroceramides are clearly up regulated ($P < 0.01$) while five out of the 14 glycosphingolipids and sphingomyelins are downregulated ($P < 0.01$) and none upregulated (Fig. 4). We included four biological replicates for each experimental condition and all samples were extracted with the same protocol. All biological replicates show the very same patterns (it is highly reproducible) and we believe that these changes are unlikely to arise from technical variation (i.e. that changes in the extraction efficiency would lead to changes in one class of hydrophobic molecules in only the four samples of one condition while others hydrophobic metabolites show the opposite trend).

We also understood that reviewer#3 was requiring further validation of the metabolomics data. First, as a control of the data generated by the metabolomics screen and the conclusions that can be drawn, we monitored the behavior of metabolites that should be altered upon manipulation of CD98hc expression, amino acids and amino acid-derived metabolites, a canonical function of CD98hc. For instance, we compared control cells (WT) to CD98hc null cells re-expressing CD98hc with high level of expression. We observe that metabolites in amino acid metabolism are statistically significantly reduced in control cells as compared to CD98hc overexpressing cells (which is expected since the more CD98hc, the more amino acid transport). As a note, very little sphingolipid perturbation was observed in that comparison, again as expected.

Figure 5 - Volcano plot depicting metabolites fold changes versus p-value in WT versus CD98hc re-expressing CD98hc null cells. Pink, long chain based sphingoids, ceramides and phytoceramides; magenta, glycosphingolipids and sphingomyelins; green, amino acid metabolism.

As an additional control, we also used an additional mutant, C109S that in contrast to the C330S mutant, rescued RhoA activation after mechanical stress. We observed that C109S had opposite effects on sphingolipid metabolism (i.e. it leads to an increase in sphingomyelin and glycosylated ceramides, and a decrease in 3-ketosphingosine, sphinganine, and N-palmitoyl-sphinganine) probably because of overexpression. Here again the results are highly reproducible and also very consistent. We report not only changes in one metabolite, but also a consistent metabolic pattern that cannot be interpreted by technical variability.

Figure 6 - Volcano plot depicting metabolites fold changes versus p-value in C109S versus CD98hc re-expressing CD98hc null cells. Pink, long chain based sphingoids, ceramides and phytoceramides; magenta, glycosphingolipids and sphingomyelin; green, amino acid metabolism.

Finally, we used an orthogonal method (a sphingomyelin content assay kit commercially available (Supp Fig 4G)) that confirmed the LC-MS/MS-based data, i.e. an overall 25% downregulation of sphingomyelins in CD98hc null cells as compared to control cells.

Also, throughout the manuscript, we carried out a number of rescue experiments by providing cells with exogenous SM which indicated that decreased sphingolipid content was the cause of defective mechanosensing.

Nevertheless, we agree with the reviewer that targeted metabolomics that include specific extraction methods for certain metabolites will result in a higher reproducibility, better sensitivity and over all a more reliable analysis. Still, metabolites that are highly regulated between two conditions should lead to the same conclusions in global metabolomics. We would like to thank the reviewer for his/her insightful suggestion, as we feel teaming up with metabolomics specialists strengthen our work and helped us validating the results.

REVIEWERS' COMMENTS:

Reviewer #1 (Remarks to the Author):

Regarding integrin dynamic, the authors have adequately responded to my comments, and I have no further questions or suggestions for improving the manuscript. The manuscript is now suitable for publication in Nat Com.

Reviewer #3 (Remarks to the Author):

Comments on NCOMMS-17-33042B.

These comments are on the metabolomics results only.

The authors efforts, to understand the metabolomics by teaming up with experts in the field, has really improved this section. I think the section is much better now.

My only comment is that the authors should adjust their significance threshold in line with the number analytes analysed, in a most conservative way that would be a Bonferroni correction, which would set the threshold for significance at $p < 0.05/447 \Rightarrow p < 0.0001$.

This does not mean that I am not convinced about the data, but I rather would like the authors to change text and would not use the word "significant" for the difference found but rather discuss these differences as trends, maybe except for 3-ketosphinganine in figure 4.

Point by point response to reviewer's comments

Reviewer #1 (Remarks to the Author):

Regarding integrin dynamic, the authors have adequately responded to my comments, and I have no further questions or suggestions for improving the manuscript. The manuscript is now suitable for publication in Nat Com.

We acknowledge the reviewer for his comments and insightful suggestions which helped improve the quality of the manuscript.

Reviewer #3 (Remarks to the Author):

Comments on NCOMMS-17-33042B.

These comments are on the metabolomics results only.

The authors efforts, to understand the metabolomics by teaming up with experts in the field, has really improved this section. I think the section is much better now.

My only comment is that the authors should adjust their significance threshold in line with the number analytes analysed, in a most conservative way that would be a Bonferroni correction, which would set the threshold for significance at $p < 0.05/447 \Rightarrow p < 0.0001$. This does not mean that I am not convinced about the data, but I rather would like the authors to change text and would not use the word "significant" for the difference found but rather discuss these differences as trends, maybe except for 3-ketosphinganine in figure 4.

We acknowledge the reviewer for his comments. We agree that we did not correct for multiple testing and therefore performed the well-accepted Benjamini-Hochberg procedure with the false discovery rate (FDR) set to 5% to control Type I errors. These results are now reported in Supplementary Data File 1 and the calculation is described in detail in the methods section.

As there are many different opinions, on which method to use for which experiments, we briefly describe the reasons why we have chosen an FDR based approach. The goal of correcting for multiple testing is to control Type I errors. In the most conservative way one would control the probability of at least one Type I error. The Bonferroni correction (the most conservative as also mentioned by the reviewer) or the Holm-Bonferroni method are the most common methods to be used, if a single false positive would be critical. In cases where the number of tests is large, such conservative methods reject a lot of true positives in order to control the probability of at least one Type I error. False discovery rate (FDR) based procedures are then often the method of choice, as they have a greater power. In our case, a very small number of Type I errors can be tolerated, as the output underlies further criteria like being associated with the same metabolic pathway or being validated with an orthogonal method. In order to keep the number of Type I errors small, we have chosen a quite stringent FDR of 5%

We thank the reviewer for his suggestion to rethink the use of the word "significant" in the context of the metabolomics results. In the manuscript, we use the metabolomics results in a defined context, and thus the corresponding statistical tests. A reader might want to use the metabolomics results in a different context, which would require other significance levels. We thus avoid the word significant in the text as suggested by the reviewer. We now

provide and refer to actual statistical results including FDR, in order for the reader to be able to critically assess our data and make up his/her own mind.